# Ancient human genome-wide data from a 3000-year interval in the Caucasus corresponds with eco-geographic regions

Chuan-Chao Wang et al.[#]

Archaeogenetic studies have described the formation of Eurasian 'steppe ancestry' as a mixture of Eastern and Caucasus hunter-gatherers. However, it remains unclear when and where this ancestry arose and whether it was related to a horizon of cultural innovations in the 4th millennium BCE that subsequently facilitated the advance of pastoral societies in Eurasia. Here we generated genome-wide SNP data from 45 prehistoric individuals along a 3000-year temporal transect in the North Caucasus. We observe a genetic separation between the groups of the Caucasus and those of the adjacent steppe. The northern Caucasus groups are genetically similar to contemporaneous populations south of it, suggesting human movement across the mountain range during the Bronze Age. The steppe groups from Yamnaya and subsequent pastoralist cultures show evidence for previously undetected farmer-related ancestry from different contact zones, while Steppe Maykop individuals harbour additional Upper Palaeolithic Siberian and Native American related ancestry.

. Correspondence and requests for materials should be addressed to C.-C.W. (email: wang@xmu.edu.cn) or to S.H. (email: svend.hansen@dainst.de) or to J.K. (email: krause@shh.mpg.de) or to W.H. (email: haak@shh.mpg.de). [#]A full list of authors and their affiliations appears at the end of the paper.

The 1100-kilometre long Caucasus mountain ranges extend between the Black Sea and the Caspian Sea and are bounded by the rivers Kuban and Terek in the north and the Kura and Araxes rivers in the south (Fig. 1). The rich archaeological record suggests extensive human occupation since the Upper Palaeolithic[1–3]. A Neolithic lifestyle based on food production began in the Caucasus after 6000 calBCE[4]. As a region rich in natural resources such as ores, pastures and timber, the Caucasus gained increasing importance to the economies of the growing urban centres in northern Mesopotamia[5,6]. In the 4th millennium BCE the archaeological record attests to the presence of the Maykop and Kura-Araxes, two major cultural complexes of the Bronze Age (BA) in the region (Fig. 1, Supplementary Note 1). The Maykop culture is well known for its large and rich burial mounds, especially at the eponymous Maykop site, which reflect the rise of a new system of social organization[7], while the Kura-Araxes is found on both flanks of the Caucasus mountain range, demonstrating a connection between north and south[5].

Contact between the near East, the Caucasus, the Steppe and central Europe is documented, both archaeologically and genetically, as early as the 5th millennium BC[8–10]. This increased in the 4th millennium BCE along with the development of new technologies such as the wheel and wagon, copper alloys, new weaponry, and new breeds of domestic sheep[11]. Such contact was critical in the cultural[12] and genetic formation of the Yamnaya complex on the Eurasian Steppe—with about half of BA Steppe ancestry thought to derive from the Caucasus[13]. In the 3rd millennium BC, increased mobility associated with wheeled transport and the intensification of pastoralist practices led to dramatic expansions of populations closely related to the Yamnaya[14–16], accompanied by the domestication of horses[17] allowing more efficient keeping of larger herds. These expansions ultimately contributed a substantial fraction to the ancestry of present-day Europe and South Asia[18–20]. Thus, the Caucasus region played a crucial role in the prehistory and formation of Eurasian genetic diversity.

Recent ancient DNA studies have resolved several long-standing questions regarding cultural and population transformations in prehistory. One important feature is a cline of European hunter-gatherer (HG) ancestry that runs roughly from West to East (hence WHG and EHG; blue component in Fig. 2a,c). This ancestry differs from that of Early European farmers, who are more closely related to farmers of northwest Anatolia[21,22] and also to pre-farming Levantine individuals[9]. The near East and Anatolia have long-been seen as the regions from which European farming and animal husbandry emerged. In the Mesolithic and Early Neolithic, these regions harboured three divergent populations, with Anatolian and Levantine ancestry in the west, and a group with a distinct ancestry in the east. The latter was first described in Upper Pleistocene individuals from Georgia (Caucasus hunter-gatherers; CHG)[13] and then in Mesolithic and Neolithic individuals from Iran[9,23]. The following millennia, spanning the Neolithic to BA, saw admixture between these ancestral groups, leading to a pattern of genetic homogenization of the source populations[9]. North of the Caucasus, Eneolithic and BA individuals from the Samara region (5200–4000 BCE) carry an equal mixture of EHG- and CHG/Iranian ancestry, so-called 'steppe ancestry'[13] that eventually spread further west[18,19], where it contributed substantially to present-day Europeans, and east to the Altai region as well as to South Asia[9].

To understand and characterize the genetic variation of Caucasian populations, present-day groups from various geographic, cultural/ethnic and linguistic backgrounds have been analyzed previously[24–26]. Yunusbayev and colleagues described the Caucasus region as an asymmetric semipermeable barrier based on a higher genetic affinity of southern Caucasus groups to Anatolian and near Eastern populations and a genetic discontinuity between these and populations of the North Caucasus and the adjacent Eurasian steppes. While autosomal and mitochondrial DNA data appear relatively homogeneous across the entire Caucasus, the Y-chromosome diversity reveals a deeper genetic structure attesting to several male founder effects, with striking correspondence to geography, ethnic and linguistic groups, and historical events[24,25].

In our study, we aimed to investigate when and how the genetic patterns observed today were formed and test whether they have been present since prehistoric times by generating time-stamped human genome-wide data. We were also interested in characterizing the role of the Caucasus as a conduit for gene-flow in the past and in shaping the cultural and genetic makeup of the wider region (Supplementary Note 1). This has important implications for understanding the means by which Europe, the Eurasian steppe zone, and the earliest urban centres in the Near East were connected[6]. We aimed to genetically characterise individuals from cultural complexes such as the Maykop and Kura-Araxes and assessing the amount of gene flow in the Caucasus during times when the exploitation of resources of the steppe environment intensified, since this was potentially triggered by the cultural and technological innovations of the Late Chalcolithic and EBA around 4000–3000 BCE[5] (Supplementary Note 2). Finally, since the spread of steppe ancestry into central Europe and the eastern steppes during the early 3rd millennium BCE was a striking migratory event in human prehistory[18,19], we also retraced the formation of the steppe ancestry profile and tested for influences from neighbouring farming groups to the west or early urbanization centres further south.

Here we show that individuals from our Caucasian time transect form two distinct genetic clusters that were stable over 3000 years and correspond with eco-geographic zones of the steppe and mountain regions. This finding is different from the situation today, where the Caucasus mountains separate northern from southern Caucasus populations. However, during the early BA we also observe subtle gene flow from the Caucasus as well as the eastern European farming groups into the steppe region, which predates the massive expansion of the steppe pastoralists that followed in the 3rd millennium BCE[18,19].

## Results

**Genetic clustering and uniparentally inherited markers**. We report genome-wide data at a targeted set of 1.2 million single nucleotide polymorphisms (SNPs)[18,27] for 59 Eneolithic and BA individuals from the Caucasus region. After filtering out 14 individuals that were first-degree relatives or showed evidence of contamination (Supplementary Data 1, Supplementary Note 3) we retained 45 individuals for downstream analyses using a cutoff of 30,000 SNPs. We merged our newly generated samples with previously published ancient and modern data (Supplementary Data 2). We first performed principal component analysis (PCA)[28] and ADMIXTURE[29] analysis to assess the genetic affinities of the ancient individuals qualitatively (Fig. 2). Based on PCA and ADMIXTURE plots we observe two distinct genetic clusters: one falls with previously published ancient individuals from the West Eurasian steppe (hence termed 'Steppe'), and the second clusters with present-day southern Caucasian populations and ancient BA individuals from today's Armenia (henceforth called 'Caucasus'), while a few individuals take on intermediate positions between the two. The stark distinction seen in our temporal transect is also visible in the Y-chromosome haplogroup distribution, with R1/R1b1 and Q1a2 types in the Steppe and L, J, and G2 types in the Caucasus cluster (Fig. 3a, Supplementary Data 1, Supplementary Note 4). In contrast, the mitochondrial haplogroup distribution is more diverse and similar in both groups (Fig. 3b, Supplementary Data 1).

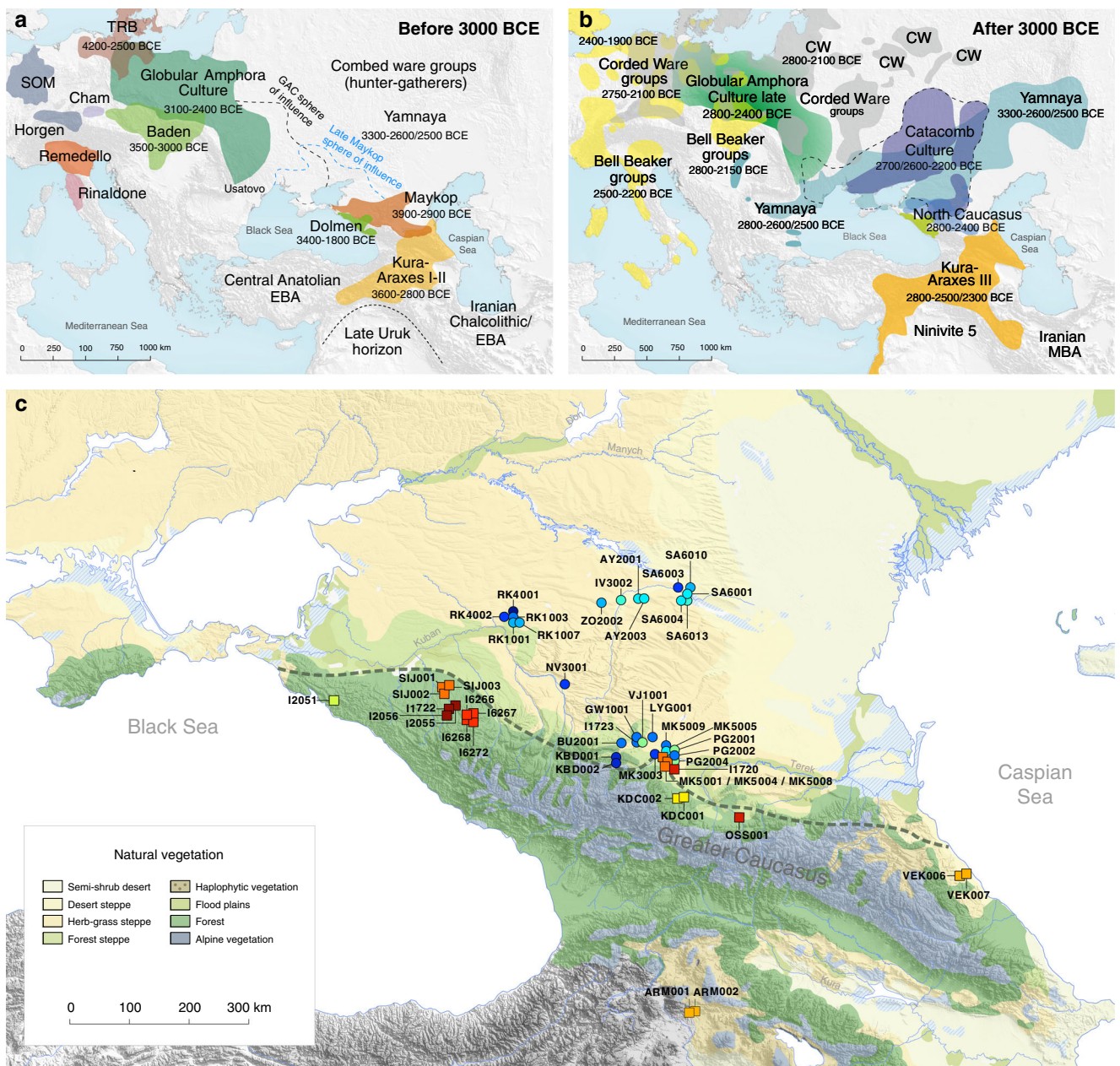

**Fig. 1** Map of samples, sites and archaeological cultures mentioned in this study. Temporal and geographic distribution of archaeological cultures is shown for two windows in time **a**, **b** that are critical for our data. The zoomed map **c** shows the location of studied individuals from various sites in the Caucasus. Symbols and sample names correspond with Fig. 2 and Supplementary Data 1. The dashed line illustrates a hypothetical geographic border between genetically distinct *Steppe* and *Caucasus* clusters. (BB Bell Beaker; CW Corded Ware; TRB Trichterbecher/Funnel Beaker; SOM Seine-Oise-Marne complex). (All three maps were prepared by S. Reinhold and D. Mariaschk based on freely available geological and vegetation GIS-data from https://www.usgs.gov/, https://www.naturalearthdata.com/ and modified after Stone, T.A., and P. Schlesinger. 2003. RLC Vegetative Cover of the Former Soviet Union, 1990. ORNL DAAC, Oak Ridge, Tennessee, USA. https://doi.org/10.3334/ORNLDAAC/700.)

The two distinct clusters are already visible in the oldest individuals of our temporal transect, dated to the Eneolithic period (~6300–6100 yBP/4300–4100 calBCE). Three individuals from the sites of Progress 2 and Vonyuchka 1 in the North Caucasus piedmont steppe ('Eneolithic steppe'), which harbour EHG and CHG related ancestry, are genetically very similar to Eneolithic individuals from Khvalynsk II and the Samara region[18,22]. This extends the cline of dilution of EHG ancestry via CHG-related ancestry to sites immediately north of the Caucasus foothills (Fig. 1c; Fig. 2d).

In contrast, the oldest individuals from the northern mountain flank itself, which are three first-degree-related individuals from the Unakozovskaya cave associated with the Darkveti-Meshoko Eneolithic culture (analysis label 'Eneolithic Caucasus') show mixed ancestry mostly derived from sources related to the Anatolian Neolithic (orange) and CHG/Iran Neolithic (green) in the ADMIXTURE plot (Fig. 2c). While similar ancestry profiles have been reported for Anatolian and Armenian Chalcolithic and BA individuals[9,19], this result suggests the presence of this mixed ancestry north of the Caucasus as early as ~6500 years ago.

**Ancient North Eurasian ancestry in Steppe Maykop individuals.** Four individuals from mounds in the grass steppe zone, archaeologically associated with the 'Steppe Maykop' cultural

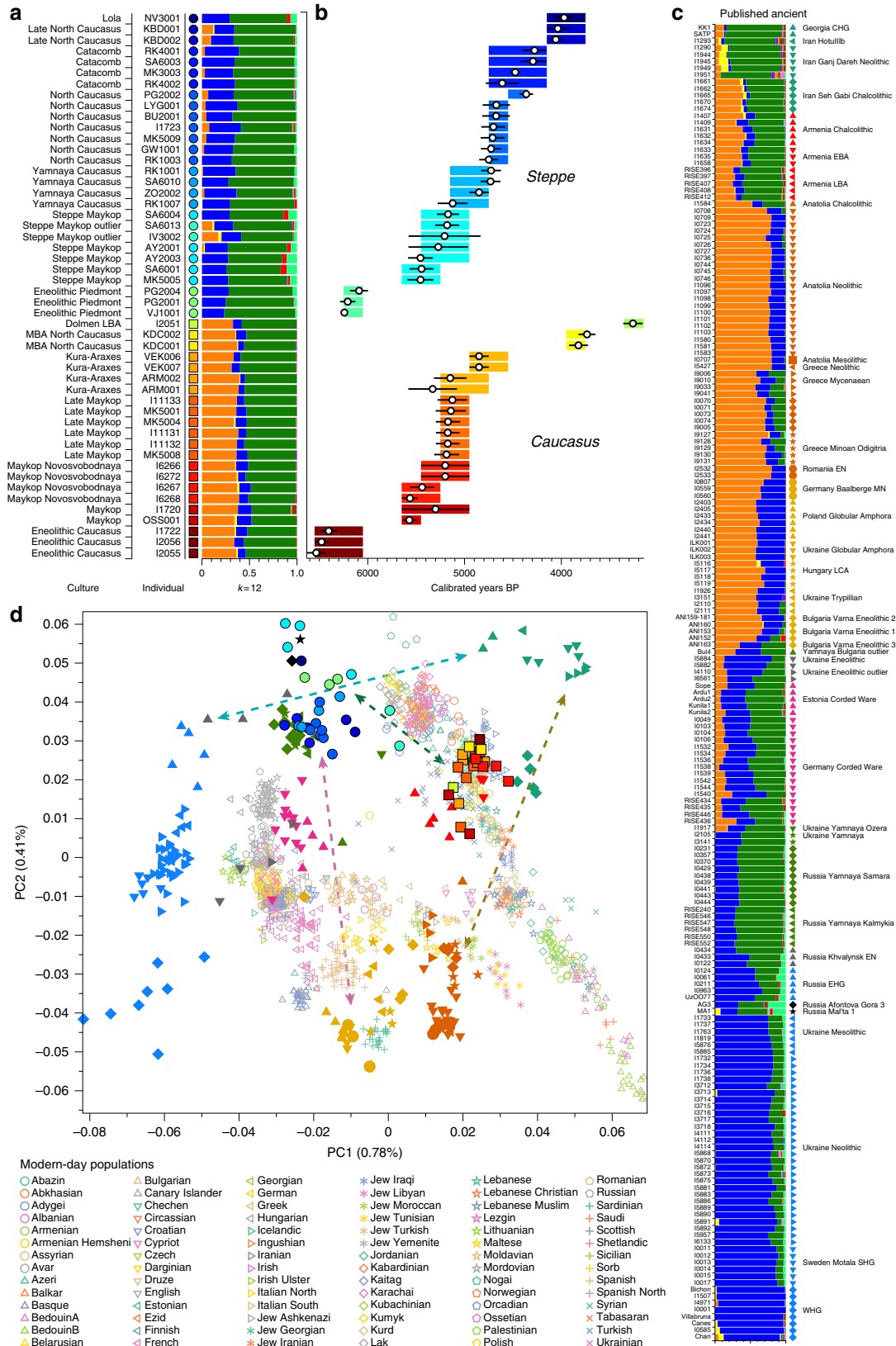

**Fig. 2** ADMIXTURE and PCA results, and chronological order of ancient Caucasus individuals. **a** *ADMIXTURE* results ($k = 12$) of the newly genotyped individuals (filled symbols with black outlines) sorted by genetic clusters (*Steppe* and *Caucasus*) and in chronological order (coloured bars indicate the relative archaeological dates, **b** white circles the mean calibrated radiocarbon date and the errors bars the 2-sigma range. **c** *ADMIXTURE* results of relevant prehistoric individuals mentioned in the text (filled symbols), and **d** shows these projected onto a PCA of 84 modern-day West Eurasian populations (open symbols). Dashed arrows indicate trajectories of admixture: EHG—CHG (petrol), Yamnaya—Central European MN (pink), Steppe—Caucasus (green), and Iran Neolithic—Anatolian Neolithic (brown)

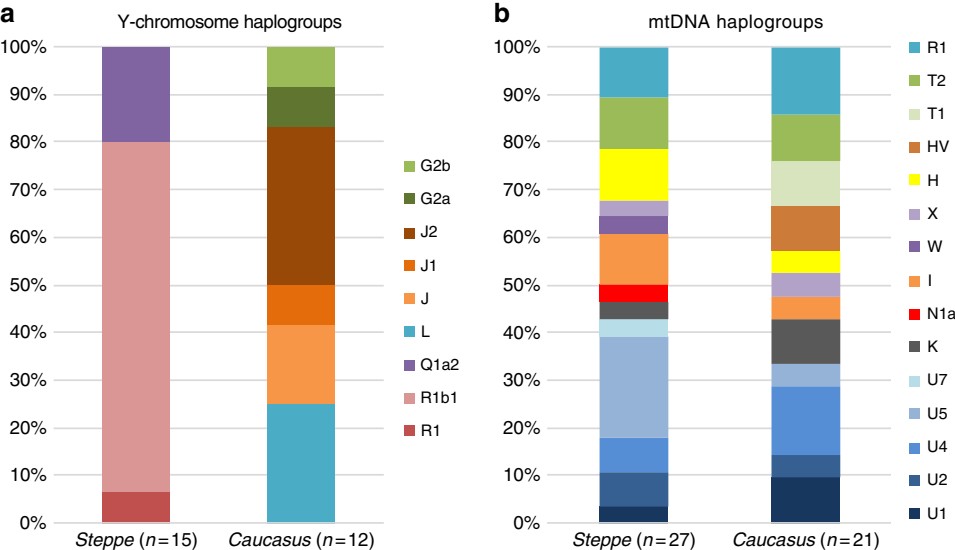

**Fig. 3** Results from uniparentally inherited markers. Comparison of Y-chromosome **a** and mitochondrial haplogroup **b** distribution in the *Steppe* and *Caucasus* cluster

complex (Supplementary Notes 1 and 2), lack the Anatolian farmer-related (AF) component when compared to contemporaneous Maykop individuals from the foothills. Instead they carry a third and fourth ancestry component that is linked deeply to Upper Paleolithic Siberians (maximized in the individual Afontova Gora 3 (AG3)[30,31] and Native Americans, respectively, and in modern-day North Asians, such as North Siberian Nganasan (Supplementary Data 3). To illustrate this affinity with 'ancient North Eurasians' (ANE)[21], we also ran PCA with 147 Eurasian (Supplementary Fig. 1A) and 29 Native American populations (Supplementary Fig. 1B). The latter represents a cline from ANE-rich steppe populations such as EHG, Eneolithic individuals, AG3 and Mal'ta 1 (MA1) to modern-day Native Americans at the opposite end. To formally test the excess of alleles shared with ANE/Native Americans we performed $f_4$-statistics of the form $f_4$(Mbuti, X; Steppe Maykop, Eneolithic steppe), which resulted in significantly positive Z-scores (Z >3) for AG3, MA1, EHG, Clovis and Kennewick for the ancient populations and many present-day Native American populations (Supplementary Table 1). Based on these observations we used qpWave and qpAdm methods to model the number of ancestral sources contributing to the Steppe Maykop individuals and their relative ancestry coefficients. Simple two-way models of Steppe Maykop as an admixture of Eneolithic steppe, AG3 or Kennewick do not fit (Supplementary Table 2). However, we could successfully model Steppe Maykop ancestry as being derived from populations related to all three sources (p-value 0.371 for rank 2): Eneolithic steppe (63.5 ± 2.9%), AG3 (29.6 ± 3.4%) and Kennewick (6.9 ± 1.0%) (Fig. 4; Supplementary Table 3). We note that the Kennewick related signal is most likely driven by the East Eurasian part of Native American ancestry as the $f_4$-statistics (Steppe_Maykop, Fitted Steppe_Maykop; Outgroup1, Outgroup2) show that the Steppe Maykop individuals share more alleles not only with Karitiana but also with Han Chinese (Supplementary Table 2).

**Characterising the *Caucasus* ancestry profile.** The Maykop period, represented by 12 individuals from eight Maykop sites (Maykop, n = 2; a cultural variant 'Novosvobodnaya' from the site Klady, n = 4; and Late Maykop, n = 6) in the northern foothills appears homogeneous. These individuals closely resemble the preceding Eneolithic Caucasus individuals and present a

continuation of the local genetic profile. This ancestry persists in the following centuries at least until ~3100 yBP (1100 calBCE), as revealed by individuals from Kura-Araxes from both the northeast (Velikent, Dagestan) and the South Caucasus (Kaps, Armenia), as well as MBA/LBA individuals (e.g. Kudachurt, Marchenkova Gora) from the north. Overall, this *Caucasus* ancestry profile falls among the 'Armenian and Iranian Chalcolithic' individuals and is indistinguishable from other Kura-Araxes individuals (Armenian EBA) on the PCA plot (Fig. 2), suggesting a dual origin involving Anatolian/Levantine and Iran Neolithic/CHG ancestry, with only minimal EHG/WHG contribution possibly as part of the AF ancestry[9].

Admixture $f_3$-statistics of the form $f_3$(X, Y; target) with the *Caucasus* cluster as target resulted in significantly negative Z scores (Z < −3) when CHG (or AG3 in Late Maykop) were used as one and Anatolian farmers as the second potential source (Supplementary Table 4). We also used qpWave to determine the number of streams of ancestry and found that a minimum of two is sufficient (Supplementary Table 5).

We then tested whether each temporal/cultural group of the *Caucasus* cluster could be modelled as a simple two-way admixture by exploring all possible pairs of sources in qpWave. We found support for CHG as one source and AF ancestry or a derived form such as is found in southeastern Europe as the other (Supplementary Table 6). We focused on mixture models of proximal sources (Fig. 4b) such as CHG and Anatolian Chalcolithic for all six groups of the *Caucasus* cluster (Eneolithic Caucasus, Maykop and Late Makyop, Maykop-Novosvobodnaya, Kura-Araxes, and Dolmen LBA), with admixture proportions on a genetic cline of 40–72% Anatolian Chalcolithic related and 28–60% CHG related (Supplementary Table 7). When we explored Romania_EN and Bulgaria_Neolithic individuals as alternative southeast European sources (30–46% and 32–49%), the CHG proportions increased to 54–70% and 51–68%, respectively. We hypothesize that alternative models, replacing the Anatolian Chalcolithic individual with yet unsampled populations from eastern Anatolia, South Caucasus or northern Mesopotamia, will likely also provide a fit to some of the tested *Caucasus* groups. Models with Iran Neolithic as substitute for CHG could also explain the data in a two-way admixture with the combination of Armenia Chalcolithic or Anatolia Chalcolithic as the other source. However, models replacing CHG with EHG

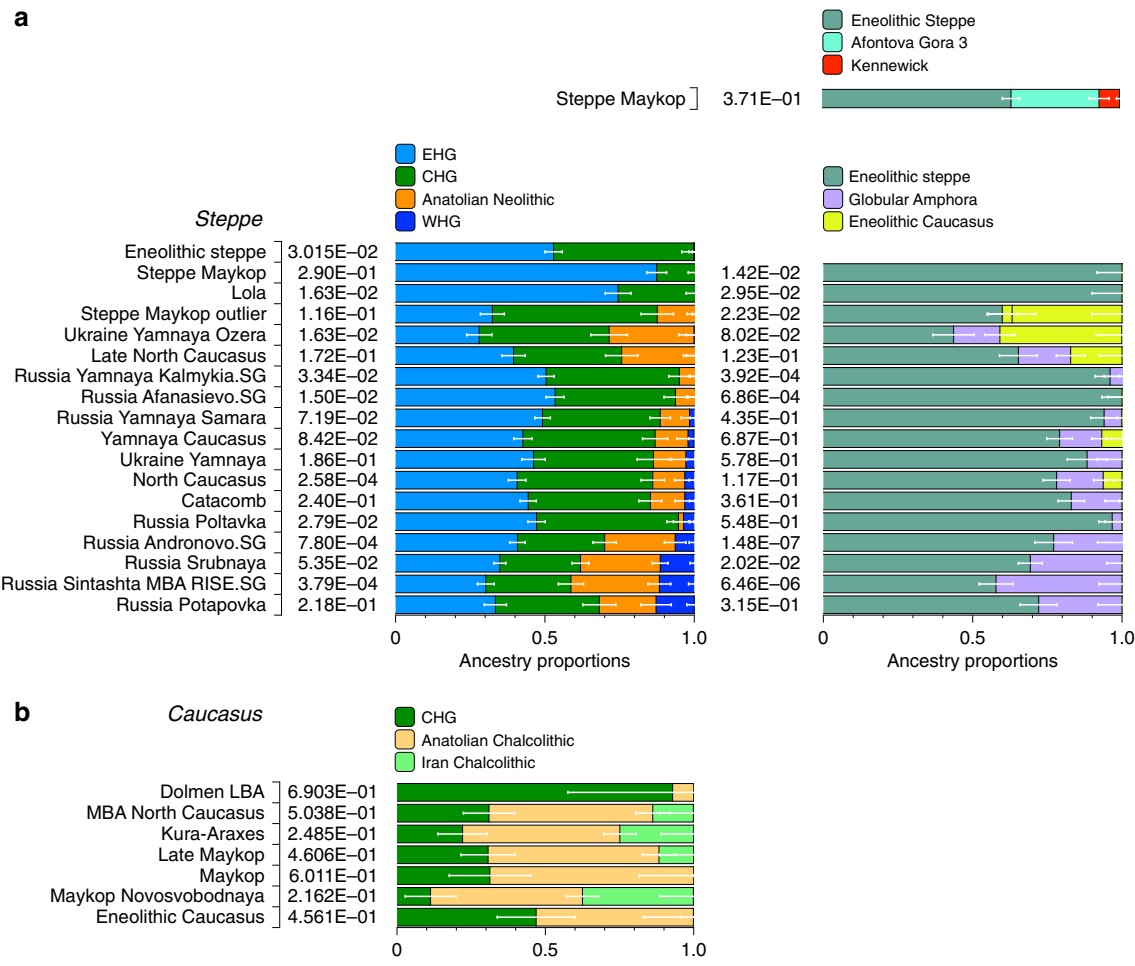

**Fig. 4** Modelling results for the Steppe and Caucasus cluster. Admixture proportions based on (temporally and geographically) distal and proximal models, showing additional AF ancestry in Steppe groups (**a**) and additional gene flow from the south in some of the Steppe groups as well as the Caucasus groups (**b**) (see also Supplementary Tables 10, 14 and 19)

received no support (Supplementary Table 8), indicating no strong influence for admixture from the adjacent steppe to the north. We also found no direct evidence of EHG or WHG ancestry in *Caucasus* groups (Supplementary Table 9), but observed that Kura-Araxes and Maykop-Novosvobodnaya individuals had likely received additional Iran Chalcolithic-related ancestry (24.9% and 37.4%, respectively; Fig. 4; Supplementary Table 10).

**Characterising the *Steppe* ancestry profile**. Individuals from the North Caucasian steppe associated with the Yamnaya cultural formation (5300–4400 BP, 3300–2400 calBCE) appear genetically almost identical to previously reported Yamnaya individuals from Kalmykia[19] immediately to the north, the middle Volga region[18,22], Ukraine, and to other BA individuals from the Eurasian steppes who share the characteristic 'steppe ancestry' profile as a mixture of EHG and CHG-related ancestry[9,13]. These individuals form a tight cluster in PCA space (Fig. 2) and can be shown formally to be a mixture by significantly negative admixture $f_3$-statistics of the form $f_3$(EHG, CHG; target) (Supplementary Fig. 2). This cluster also involves individuals of the North Caucasus culture (4800–4500 BP, 2800–2500 calBCE) in the piedmont steppe, who share the steppe ancestry profile, as do individuals from the Catacomb culture in the Kuban, Caspian and piedmont steppes (4600–4200 BP, 2600–2200 calBCE), which succeeded the Yamnaya horizon.

The individuals of the MBA post-Catacomb horizon (4200–3700 BP, 2200–1700 calBCE) such as Late North Caucasus and Lola cultures represent both ancestry profiles common in the North Caucasus: individuals from the mountain site Kabardinka show a typical steppe ancestry profile, whereas individuals from the site Kudachurt 90 km to the west or our most recent individual from the western LBA Dolmen culture (3400–3200 BP, 1400–1200 calBCE) retain the 'southern' Caucasus profile. In contrast, one Lola culture individual resembles the ancestry profile of the Steppe Maykop individuals.

**Admixture into the steppe zone from the south**. Evidence for interaction between the *Caucasus* and the *Steppe* clusters is visible in our genetic data from individuals associated with the later Steppe Maykop phase around 5300–5100 years ago. These 'outlier' individuals were buried in the same mounds as those with steppe and in particular Steppe Maykop ancestry profiles but share a higher proportion of AF ancestry visible in the ADMIXTURE plot and are also shifted towards the *Caucasus* cluster in PC space (Fig. 2d). This observation is confirmed by formal D-statistics (Supplementary Fig. 3). By modelling Steppe Maykop outliers successfully as a two-way mixture of Steppe Maykop and representatives of the *Caucasus* cluster (Supplementary Table 3), we can show that these individuals received additional 'Anatolian and Iranian Neolithic ancestry', most likely from contemporaneous sources in the south. We used ALDER[32]

to estimate an average admixture time for the observed farmer-related ancestry in Steppe Maykop outliers of 20 generations or 560 years ago (Supplementary Note 5).

**Anatolian farmer-related ancestry in steppe groups.** Eneolithic Samara individuals form a cline in PC space running from EHG to CHG (Fig. 2d), which is continued by the newly reported Eneolithic steppe individuals. However, the trajectory of this cline changes in the subsequent centuries. Here we observe a cline from Eneolithic_steppe towards the *Caucasus* cluster. We can qualitatively explain this 'tilting cline' by developments south of the Caucasus, where Iranian and AF ancestries continue to mix, resulting in a blend that is also observed in the *Caucasus* cluster, from where it could have spread onto the steppe. The first appearance of 'combined farmer-related ancestry' in the steppe zone is evident in Steppe Maykop outliers. However, PCA results suggest that Yamnaya and later groups of the West Eurasian steppe carry also some farmer-related ancestry as they are slightly shifted towards 'European Neolithic groups' in PC2 (Fig. 2d) compared to the preceding Eneolithic steppe individuals. The 'tilting cline' is also confirmed by admixture $f_3$-statistics, which provide statistically significant negative values for AG3 and any AF group as the two sources (Supplementary Table 11). Using $f$- and $D$-statistics we also observe an increase in farmer-related ancestry (both Anatolian and Iranian) in our *Steppe* cluster, distinguishing the Eneolithic steppe from later groups. In addition, we find the *Caucasus* cluster or Levant/AF groups to share more alleles with *Steppe* groups than with EHG or Samara_Eneolithic (Supplementary Figs. 4 and 5). MLBA groups such as Poltavka, Andronovo, Srubnaya, and Sintashta show a further increase of AF ancestry consistent with previous studies[9,22], reflecting different processes not directly related to events in the Caucasus (Supplementary Fig. 6).

We then used *qpWave* and *qpAdm* to explore the number of ancestry sources for the AF component to evaluate whether geographically proximate groups contributed plausibly to the subtle shift of Eneolithic ancestry in the steppe towards Neolithic groups. Specifically, we tested whether any of the Eurasian steppe ancestry groups can be successfully modelled as a two-way admixture between Eneolithic steppe and a population X derived from Anatolian- or Iranian farmer-related ancestry, respectively. Surprisingly, we found that a minimum of four streams of ancestry is needed to explain all eight steppe ancestry groups tested (Fig. 2; Supplementary Table 12). Importantly, our results show a subtle contribution of both AF ancestry and WHG-related ancestry (Fig. 4; Supplementary Tables 13 and 14), likely brought in through MN/LN farming groups from adjacent regions in the West. A direct source of AF ancestry can be ruled out (Supplementary Table 15). At present, due to the limits of our resolution, we cannot identify a single best source population. However, geographically proximal and contemporaneous groups such as Globular Amphora and Eneolithic groups from the Black Sea area (Ukraine and Bulgaria), representing all four distal sources (CHG, EHG, WHG, and Anatolian_Neolithic), are among the best supported candidates (Fig. 4; Supplementary Table 16). Applying the same method to the subsequent North Caucasian *Steppe* groups such as Catacomb, (Late) North Caucasus confirms this pattern (Supplementary Table 16).

Using *qpAdm* with Globular Amphora as a proximate surrogate population, we estimated the contribution of AF ancestry into Yamnaya and other steppe groups. We find that Yamnaya Samara individuals have $13.2 \pm 2.7\%$ and Ukraine or Caucasus Yamnaya individuals $16.6 \pm 2.9\%$ AF ancestry (Fig. 4; Supplementary Table 17)—statistically indistinguishable proportions. Substituting Globular Amphora with Iberia Chalcolithic

does not alter the results profoundly (Supplementary Table 18). This suggests that the source population was a mixture of AF ancestry and a minimum of 20% WHG ancestry, a genetic profile shared by many European MN/LN and Chalcolithic individuals of the 3rd millennium BCE analysed thus far.

To account for potentially un-modelled ancestry from the *Caucasus* groups, we added 'Eneolithic Caucasus' as an additional source to build a three-way model. We found that Yamnaya Caucasus, Yamnaya Ukraine Ozera, North Caucasus and Late North Caucasus had likely received additional ancestry (6–40%) from nearby *Caucasus* groups (Supplementary Table 19). This suggests a more complex and dynamic picture of steppe ancestry groups through time, including the formation of a local variant of steppe ancestry in the North Caucasian steppe from the local Eneolithic, a contribution of Steppe Maykop groups, and population continuity between the early Yamnaya period and the MBA (5300–3200 BP, 3300–2200 calBCE).

**Insights from micro-transects through time.** The availability of multiple individuals from one burial mounds allowed us to test genetic continuity on a micro-transect level. By focusing on two kurgans (Marinskaya 5 and Sharakhalsun 6) with four and five individuals, respectively, we observe that the genetic ancestry varied through time, alternating between the *Steppe* and *Caucasus* ancestries (Supplementary Fig. 7), suggesting a shifting genetic border between the two genetic clusters. We also detected various degrees of kinship between individuals buried in the same mound, which supports the view that particular mounds reflected genealogical lineages. Overall, we observe a balanced sex ratio within our sites across the individuals tested (Supplementary Note 4).

**A joint model of ancient populations of the Caucasus region.** Our fitted *qpGraph* model recapitulates the genetic separation between the *Caucasus* and *Steppe* groups with the Eneolithic steppe individuals deriving more than 60% of ancestry from EHG and the remainder from a CHG-related basal lineage, whereas the Maykop group received about 86.4% from CHG, 9.6% Anatolian farming related ancestry, and 4% from EHG. The Yamnaya individuals from the Caucasus derived the majority of their ancestry from Eneolithic steppe individuals, but also received about 16% from Globular Amphora-related farmers (Fig. 5, Supplementary Note 6).

**Discussion**
Our data from the Caucasus region cover a 3000-year interval of prehistory, during which we observe a genetic separation between the groups in the northern foothills and those groups of the bordering steppe regions in the north (i.e. the 'real' steppe). We have summarised these broadly as *Caucasus* and *Steppe* groups in correspondence with eco-geographic vegetation zones that characterise the socio-economic basis of the associated archaeological cultures.

When compared to present-day human populations from the Caucasus, which show a clear separation into North and South Caucasus groups along the Great Caucasus mountain range (Fig. 2d), our new data highlight a different situation during the BA. The fact that individuals buried in kurgans in the North Caucasian piedmont zone are more closely related to ancient individuals from regions further south in today's Armenia, Georgia and Iran results in two main observations.

First, sometime after the BA present-day North Caucasian populations must have received additional gene-flow from steppe populations that now separates them from southern Caucasians, who largely retained the BA ancestry profile. The archaeological

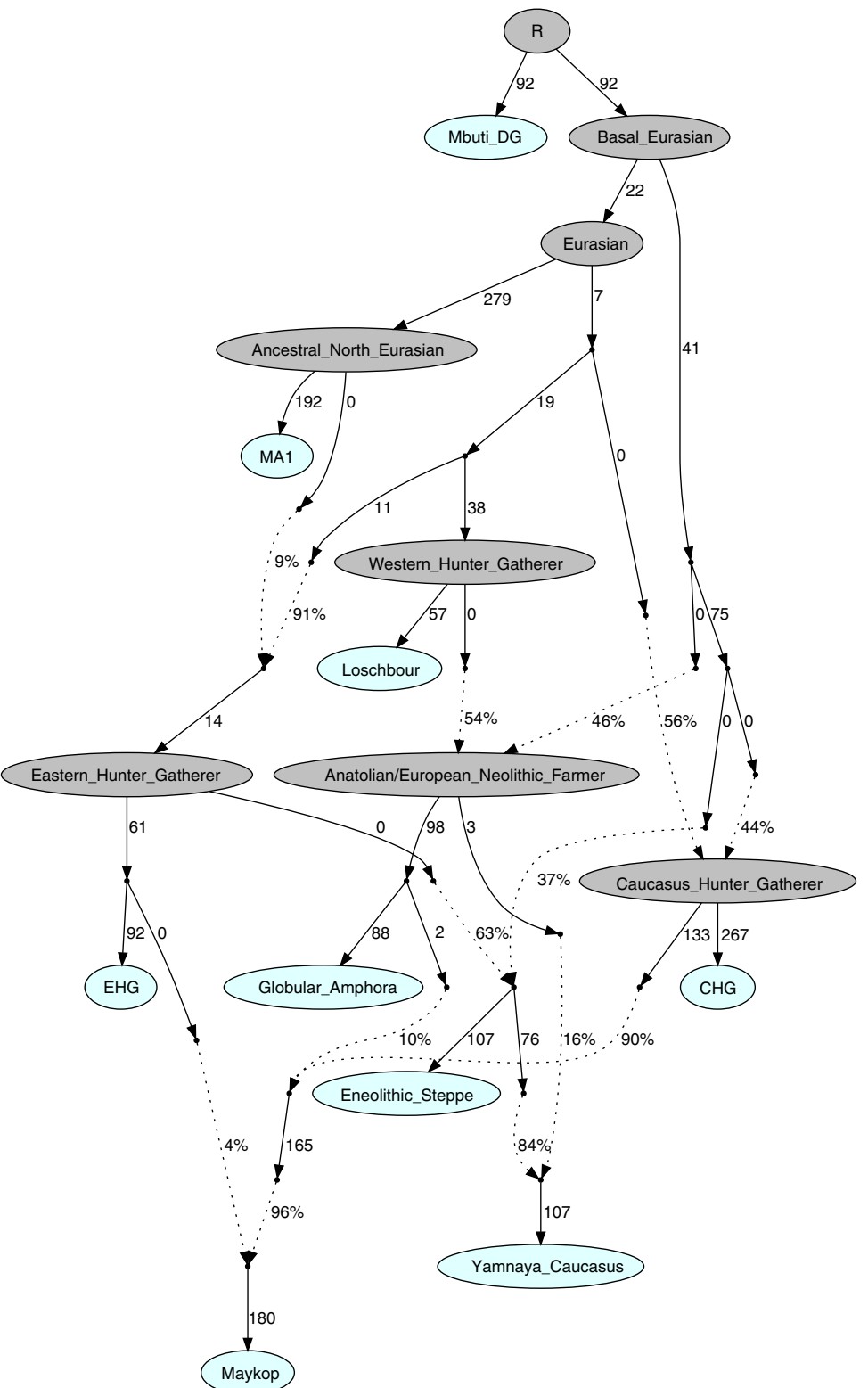

**Fig. 5** Admixture Graph modelling of the population history of the Caucasus region. We started with a skeleton tree without admixture, including Mbuti, Loschbour and MA1. We grafted onto this EHG, CHG, Globular_Amphora, Eneolithic_steppe, Maykop, and Yamnaya_Caucasus, adding them consecutively to all possible edges in the tree and retaining only graph solutions that provided no differences of |Z| > 3 between fitted and estimated statistics. The lowest Z-score for this graph is |Z| = 2.824. We note that the maximum discrepancy is $f_4$(MA1, Maykop; EHG, Eneolithic_steppe) = −3.369 if we do not add the 4% EHG ancestry to Maykop. Drifts along edges are multiplied by 1000 and dashed lines represent admixture

and historic records suggest numerous incursions during the subsequent Iron Age and Medieval times[33], but ancient DNA from these time periods will be needed to test this directly.

Second, our results reveal that the Caucasus was no barrier to human movement in prehistory. Instead the interface of the steppe and northern mountain ecozones could be seen as a transfer zone of cultural innovations from the south and the adjacent Eurasian steppes to the north (Supplementary Note 1). The latter is best exemplified by the two Steppe Maykop outlier individuals, which carry additional AF ancestry, for which the contemporaneous piedmont Maykop individuals present likely candidates for the source of this ancestry. This might also explain the regular presence of 'Maykop-style artefacts' in burials that share Steppe Eneolithic traditions and are genetically assigned to the *Steppe* group. Hence the diverse 'Steppe Maykop' group indeed represents the mutual entanglement of *Steppe* and *Caucasus* groups and their cultural affiliations in this interaction sphere.

Concerning the influences from the south, our oldest dates from the immediate Maykop predecessors Darkveti-Meshoko (Eneolithic Caucasus) indicate that the *Caucasus* genetic profile was present north of the range ~6500 BP, 4500 calBCE. This is in accordance with the Neolithization of the Caucasus, which had started in the flood plains of South Caucasian rivers in the 6th millennium BCE, from where it spread across to the West/Northwest during the following millennium[4,34]. It remains unclear whether the local CHG ancestry profile (Kotias Klde and Satsurblia in today's Georgia) was also present in the North Caucasus region before the Neolithic. However, if we take the CHG ancestry as a local baseline and the oldest Eneolithic Caucasus individuals from our transect as a proxy for the local Late Neolithic ancestry, we notice a substantial increase in AF ancestry. This in all likelihood reflects the process of Neolithization, which also brought this type of ancestry to Europe. As a consequence, it is possible that Neolithic groups could have reached the northern foothills earlier[35] (Supplementary Note 1). Hence, additional sampling from older individuals would be desirable to fill this temporal and spatial gap.

We show that the North Caucasus piedmont region was genetically connected to the south at the time of the eponymous grave mound of Maykop. Even without direct ancient DNA data from northern Mesopotamia, our results suggest an increased assimilation of Chalcolithic individuals from Iran, Anatolia, and Armenia and those of the Eneolithic Caucasus during 6000–4000 calBCE[9], and thus likely also intensified cultural connections. It is possible that the cultural and genetic basis of Maykop were formed within this sphere of interaction (Fig. 4; Supplementary Table 10). In fact, the Maykop phenomenon was long understood as the terminus of expanding Mesopotamian civilisations[5,7,36]. It has been further suggested that along with these influences the key technological innovations in western Asia that had revolutionised the late 4th millennium BCE had ultimately also spread to Europe[37]. An earlier connection in the late 5th millennium BCE, however, allows speculations about an alternative archaeological scenario: was the cultural exchange mutual and did e.g. metal rich areas such as the Caucasus contribute substantially to the development and transfer of these innovations[12,38]?

Within the 3000-year interval covered in this study, we observe a degree of genetic continuity within each cluster, albeit occasionally interspersed by subtle gene-flow between the two clusters as well as from outside sources. Moreover, our data show that the northern flanks were consistently linked to the Near East and had received multiple streams of gene flow from the south during the Maykop, Kura-Araxes, and late phase of the North Caucasus culture. Interestingly, this renewed appearance of the southern genetic make-up in the foothills corresponds to a period of climatic deterioration (known as 4.2 ky event) in the steppe zone, that put a halt to the exploitation of the steppe zone for several hundred years[39]. Further insight arises from individuals that were buried in the same kurgan but in different time periods, as highlighted in the two kurgans Marinskaya 5 and Sharakhalsun 6. Here, we recognize that the distinction between *Steppe* and *Caucasus* (Fig. 1) is not strict but rather reflects a shifting border of genetic ancestry through time, possibly due to climatic/vegetation shifts and/or cultural factors linked to subsistence strategies or social exchange. Thus, the occurrence of *Steppe* ancestry in the northern foothills likely coincides with the range expansion of Yamnaya pastoralists. However, more time-stamped data from this region will be needed to provide details on the dynamics of this contact zone.

An important observation is that Eneolithic Samara and Eneolithic steppe individuals directly north of the Caucasus had initially not received AF gene flow. Instead, the Eneolithic steppe ancestry profile shows an even mixture of EHG- and CHG ancestry, suggesting an effective cultural and genetic border between the contemporaneous Eneolithic populations, notably *Steppe* and *Caucasus*. Due to the temporal limitations of our dataset, we currently cannot determine whether this ancestry is stemming from an existing natural genetic gradient running from EHG far to the north to CHG/Iran in the south or whether this is the result of Iranian/CHG-related ancestry reaching the steppe zone independently and prior to a stream of AF ancestry, where they mixed with local hunter-gatherers that carried only EHG ancestry.

All later steppe groups, starting with Yamnaya, deviate from the EHG-CHG admixture cline towards European populations in the West. We show that these individuals had received AF ancestry, in line with published evidence from Yamnaya individuals from Ukraine (Ozera) and Bulgaria[10]. In the North Caucasus, this genetic contribution could have occurred through immediate contact with *Caucasus* groups or further south. An alternative source, explaining the increase in WHG-related ancestry, would be contact with contemporaneous Chalcolithic/EBA farming groups at the western periphery of the Yamnaya distribution area, such as Globular Amphora and Cucuteni–Trypillia from Ukraine, which have been shown to carry AF ancestry[10].

Archaeological arguments are consonant with both scenarios. Contact between early Yamnaya and late Maykop groups is suggested by Maykop impulses seen in early Yamnaya complexes. A western sphere of interaction is evident from striking resemblances of imagery inside burial chambers of Central Europe and the Caucasus[40] (Supplementary Fig. 8), and similarities in geometric decoration patterns in stone cist graves in the Northern Pontic steppe[41], on stone *stelae* in the Caucasus[42], and on pottery of the Eastern Globular Amphora Culture, which links the eastern fringe of the Carpathians and the Baltic Sea[40]. This overlap of symbols implies a late 4th millennium BCE communication and interaction network that operated across the Black Sea area involving the Caucasus[43,44], and later also early Globular Amphora groups in the Carpathians and east/central Europe[45]. The role of early Yamnaya groups within this network is still unclear[41]. However, this interaction zone predates any direct influence of Yamnaya groups in Europe or the succeeding formation of the Corded Ware[46,47] and its persistence opens the possibility of subtle gene-flow from farmers at the eastern border of arable lands into the steppe, several centuries before the massive range expansions of pastoralist groups that reached Central Europe in the mid-3rd millennium BCE[18,48].

A surprising discovery was that Steppe Maykop individuals from the eastern desert steppes harboured a distinctive ancestry component that relates them to Upper Palaeolithic Siberians

(AG3, MA1) and Native Americans. This is exemplified by the more commonly East Asian features such as the derived *EDAR* allele (Supplementary Note 7), which has also been observed in HG from Karelia and Scandinavia. The additional affinity to East Asians suggests that this ancestry is not derived directly from ANE but from a yet-to-be-identified ancestral population in north-central Eurasia with a wide distribution between the Caucasus, the Ural Mountains and the Pacific coast[20], of which we have discovered the so far southwestern-most and also youngest genetic representatives.

The insight that the Caucasus mountains served as a corridor for the spread of CHG ancestry north but also for subtle later gene-flow from the south allows speculations on the postulated homelands of Proto-Indo-European (PIE) languages and documented gene-flows that could have carried a consecutive spread of both across West Eurasia[15,49]. This also opens up the possibility of a homeland of PIE south of the Caucasus, and could offer a parsimonious explanation for an early branching off of Anatolian languages, as shown on many PIE tree topologies[50–53]. Geographically conceivable are also Armenian and Greek, for which genetic data support an eastern influence from Anatolia or the southern Caucasus[10,54], and an Indo-Iranian offshoot to the east. However, latest ancient DNA results from South Asia suggest an LMBA spread via the steppe belt[20]. Irrespective of the early branching pattern, the spread of some or all of the PIE branches would have been possible via the North Pontic/Caucasus region and from there, along with pastoralist expansions, to the heart of Europe. This scenario finds support from the well attested and widely documented 'steppe ancestry' in European populations and the postulate of increasingly patrilinear societies in the wake of these expansions[48].

## Methods

**Sample collection**. Samples from archaeological human remains were collected and exported under a collaborative research agreement between the Max-Planck Institute for the Science of Human History, the German Archaeological Institute and the Lomonosov Moscow State University and Anuchin Research Institute and Museum of Anthropology (permission no. 114-18/204-03).

**Ancient DNA analysis**. We extracted DNA and prepared next-generation sequencing libraries from 107 samples in two dedicated ancient DNA laboratories at Jena and Boston, following established protocols for DNA extraction and library preparation[55,56]. Fourteen of these samples were processed at Harvard Medical School, Boston, USA, using the same protocols. Prior to sampling, all samples were irradiated with UV-light for 30 min from all sides. Teeth were sandblasted to remove the outer surface and then ground to fine powder using a mixer mill (Retsch, Germany). We also sampled the dense parts of petrous bones by cutting out a bone wedge around the region of the cochlea, which—after surface removal—was also ground to fine bone powder. We used 50–100 mg of bone powder to extract DNA. The lysis step included the addition of extraction buffer, containing 0.45M EDTA, pH 8.0, and 0.25 mg/μl Proteinase K (all Sigma-Aldrich) followed by overnight rotation at 37 °C. After centrifugation, the supernatant was transferred to a new tube, mixed with 13 ml binding buffer containing 5M GuHCl, $H_2O$ (Sigma-Aldrich), 40% Isopropanol (Merck) and 400 μl sodium acetate (Sigma-Aldrich) and then spun through silica columns (High Pure Viral Nucleic Acid Kit; Roche). The DNA bound to the columns was washed twice with 450 μl wash buffer (High Pure Viral Nucleic Acid Kit; Roche) followed by a centrifugation step at 14,000 rpm for 1 min and two dry spin steps and then eluted into a new collection tube with 100 μl TET (10 mM Tris-HCl, 1 mM EDTA, pH 8.0%, 0.1% Tween20). Blank controls were processed in parallel at a ratio of 1:7.

Double-stranded and double-indexed libraries were prepared from 25 μl DNA extract using the partial ("half") Uracil-DNA-glycosylase (UDG) protocol[56]. For initial UDG treatment we added 25 μl mastermix consisting of 0.07 U USER enzyme, 1.2X Buffer Tango (Life Technologies), 100 μM dNTP mix, 0.2 mg/ml BSA, and 1.2 mM ATP (all NEB), followed by 30 min incubation at 37 °C and 1 min at 12 °C. We then added 0.13 U UGI (Uracil Glycosylase inhibitor) and repeated the incubation step. Blunt-end-repair of the DNA fragments was carried out by adding 0.5U/T4 Polynucleotide Kinase, 0.08 U T4 DNA Polymerase (NBE), followed by incubation at 15 °C for 15 min and a standard MinElute purification step (Qiagen) eluting in 18 μl TET. Illumina adaptors (0.25 μM adapter mix) were ligated onto the blunt-ends using 1X Quick Ligase (NBE) in a total reaction volume of 40 μl, followed by another MinElute purification step. The final fill-in step included 1X isothermal buffer, 0.4 U/μl Bst-polymerase (NEB) and 125 μM dNTP

mix followed by incubation at 37 °C for 30 min, and a heat-kill step at 80 °C for 10 min. One aliquot of each library was used to quantify the DNA copy number with IS7/IS8 primers using DyNAmo SYBP Green qPCR Kit (Thermo Fisher Scientific) on the LightCycler 480 outside the clean room (Roche). Libraries were double-indexed with unique index combinations before PCR amplifications outside the cleanroom using PfuTurbo DNA Polymerase (Agilent). Indexed products were purified with MinElute columns (Qiagen) and eluted in 50 μl TET buffer and quantified with IS5/IS6 primers using the DyNAmo SYBP Green qPCR Kit (Thermo Fisher Scientific) on the LightCycler 480 (Roche). We then used Herculase II Fusion DNA Polymerase (Agilent) and IS5/IS6 primers for further amplification of the indexed products to a copy number of 10e−13 molecules/μl. After purification, the indexed libraries were quantified on a TapeStation (Agilent 4200) and pooled equimolarly to 10 nM, and then subjected to DNA sequencing (2 × 50 PE or 1 ×75 SE) on an in-house Illumina HiSeq 4000 or NextSeq 500 platform.

After initial shotgun sequencing of five million reads libraries and demultiplexing, library quality (complexity, % endogenous DNA, and DNA damage) was assessed using *EAGER*[57]. For those libraries passing quality threshholds, we carried out in-solution enrichment (1240K capture)[22] for a targeted set of 1,237,207 SNPs as well as mitochondrial genome capture, and then sequenced on for 76bp either single or paired-end. Capture sequence data were demultiplexed, adaptor clipped with leehom[58] and then further processed using *EAGER*[57], including mapping with *BWA (v0.6.1)*[59] against human genome reference GRCh37/hg19 (or just the mitochondrial reference sequence), and removing duplicate reads with the same orientation and start and end positions. To avoid an excess of remaining C-to-T and G-to-A transitions at the ends of the reads, we clipped three bases of the ends of each read for each sample using trimBam (https://genome.sph.umich.edu/wiki/BamUtil:_trimBam). We then generated pseudo-haploid calls by selecting a single read randomly for each individual at each of the targeted SNP positions using the genotype caller *pileupCaller* (https://github.com/stschiff/sequenceTools/tree/master/src-pileupCaller).

**Quality control**. We report, but have not analyzed, data from individuals that had less than 30,000 SNP hits on the 1240K set. We removed individuals with evidence of contamination based on heterozygosity in the mtDNA genome data, a high rate of heterozygosity on the X chromosome despite being male estimated with *ANGSD*[60], or an atypical ratio of the reads mapped to X versus Y chromosomes.

**Merging new and published ancient and modern population data**. We merged our newly generated ancient samples with ancient populations from the publicly available datasets (Supplementary Data 2), as well as genotyping data from worldwide modern populations using Human Origins arrays published in the same publications. We also included newly genotyped populations from the Caucasus and Asia, described in detail in Jeong et al.[61].

**Principal component analysis**. We carried out principal component analysis on Human Origins Dataset using the *smartpca* program of *EIGENSOFT*[28], using default parameters and the lsqproject: YES, numoutlieriter: 0, and shrinkmode: YES options to project ancient individuals onto the first two components.

**ADMIXTURE analysis**. We carried out *ADMIXTURE (v1.23)*[29] analysis after pruning for linkage disequilibrium in *PLINK*[62] with parameters --indep-pairwise 200 25 0.4, which retained 318,427 SNPs for the Human Origins Dataset. We ran *ADMIXTURE* with default fivefold cross-validation (--cv = 5), varying the number of ancestral populations between $K = 2$ and $K = 18$ in 100 bootstraps with different random seeds.

***f*-statistics**. We computed D-statistics and $f_4$-statistics using *qpDstat* program of *ADMIXTOOLS*[28] with default parameters. We computed the admixture $f_3$-statistics using the *qp3Pop* program of *ADMIXTOOLS* with the flag inbreed: YES. *ADMIXTOOLS* computes standard errors using the default block jackknife.

**Streams of ancestry and inference of mixture proportions**. We used *qpWave* and *qpAdm*[18] as implemented in *ADMIXTOOLS* with the option 'allsnps: YES' to test whether a set of test populations is consistent with being related via $N$ streams of ancestry from a set of outgroup populations and estimate mixture proportions for a *Test* population as a combination of $N$ 'reference' populations by exploiting (but not explicitly modeling) shared genetic drift with a set of outgroup populations: Mbuti.DG, Ust_Ishim.DG, Kostenki14, MA1, Han.DG, Papuan.DG, Onge. DG, Villabruna, Vestonice16, ElMiron, Ethiopia_4500BP.SG, Karitiana.DG, Natufian, Iran_Ganj_Dareh_Neolithic. The "DG" samples are extracted from high coverage genomes sequenced as part of the Simons Genome Diversity Project[63]. For some analyses, we used an extended set of outgroup populations, including some of the following additional ancient populations to constrain standard errors: WHG, EHG, and Levant Neolithic.

**Dating of gene-flow events**. We estimated the time depth of selected admixture events using the linkage disequilibrium (LD)-based admixture inference implemented in *ALDER*[32], assuming a generation time of 28 years[64].

**Admixture graph modelling**. Admixture graph modelling was carried out with the *qpGraph* software as implemented in *ADMIXTOOLS*[28] using Mbuti.DG as an outgroup. We explored models that jointly explain the population splits and gene flow in the Greater Caucasus region by computing $f_2$-, $f_3$- and $f_4$- statistics measuring allele sharing among pairs, triples, and quadruples of populations and evaluating fits based on the maximum $|Z|$-score comparing predicted and observed values of these statistics.

**Sex determination and Y chromosomal and mtDNA haplogroups**. We determined the sex of the newly reported samples in this study by counting the number of reads overlapping with the targets of 1240k capture reagent[31]. We extracted the reads of high base and mapping quality (samtools depth -q30 -Q37) using *samtools v1.3.1*[65]. We calculated the ratios of the numbers of reads mapped on to chromosome X or chromosome Y compared with that mapped to autosomes (X-rate and Y-rate, respectively). Samples with an X-rate < 0.42 and a Y-rate > 0.26 were assigned as males and those with an X-rate > 0.68 and a Y-rate < 0.02 were assigned as females.

We used *EAGER* and *samtools v1.3.1* to extract reads from the 1240k SNP and mitocapture data mapped to the rCRS. We used *Geneious R8.1.9*[66] to locally realign, visually inspect the pileups for contamination, and to call consensus sequences, which were used for haplotyping in *HaploGrep 2*[67]. In addition, we used the software contamMix 1.0.10, which employs a Bayesian approach to estimate contamination in the mitochondrial genome[68].

We called Y chromosomal haplogroups for males using the captured SNPs on Y chromosome by restricting to sequences with mapping quality ≥30 and bases with base quality ≥30. We determined Y chromosomal haplogroups by identifying the most derived allele upstream and the most ancestral allele downstream in the phylogenetic tree in the ISOGG version 11.89 (accessed March 31, 2016) (http://www.isogg.org/tree).

**Kinship analysis**. We used outgroup-$f_3$ statistics and the methods lcMLkin[69] and READ[70] to determine genetic kinship between individuals.

**Phenotypic SNP calls**. We determined the allele information of 5 SNPs (rs4988235, rs16891982, rs1426654, rs3827760, and rs12913832) thought to be affected by selection in our ancient samples using the captured SNPs by restricting to sequences with mapping quality ≥30 and bases with base quality ≥30 (Supplementary Note 7).

**Abbreviations**. We use the following abbreviated labels throughout the manuscript: Anatolian farmer-related AF; E Early; M Middle; L Late; N Neolithic; BA Bronze Age; WHG, EHG, CHG, HG, Western, Eastern, Caucasus hunter-gatherers, respectively; Mal'ta 1 MA1; Afontova Gora 3 AG3.

**Reporting summary**. Further information on experimental design is available in the Nature Research Reporting Summary linked to this article.

## Data availability
The aligned sequences are available through the European Nucleotide Archive under accession number PRJEB29603.

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

## Acknowledgements
We thank Stephen Clayton and all members of the MPI-SHH Archaeogenetics Department for support, Michelle O'Reilly and Hans Sell for graphics support, and Iosif Lazaridis, Vagheesh Narasimhan and Nick Patterson for critical discussions. We thank Susanne Lindauer, Ronny Friedrich, Robin van Gyseghem and Ute Blach for radiocarbon dating support. This work was funded by the Max Planck Society, the German Archaeological Institute (DAI; grant no. ERA.Net RUS, BFDJ16011), and the European Research Council (ERC) under the European Union's Horizon 2020 research and innovation programme (grant agreement n° 771234 PALEoRIDER). C.-C.W. was funded by Nanqiang Outstanding Young Talents Program of Xiamen University (X2123302), the Fundamental Research Funds for the Central Universities (ZK1144) and National Natural Science Foundation of China (31801040). E.B. and O.B. were funded by the Russian state task research for RCMG and VIGG.

## Author contributions
S.H., J.K., C.-C.W., S.R., and W.H. conceived the idea for the study design. A.W., G.B., O.C., M.F., E.H., D.K., S.M., N.R., K.S., and W.H. performed and supervised wet and dry lab work. S.H., A.K., A.R.K., V.E.M., V.G.P., V.R.E., B.C.A., R.G.M., P.L.K., K.W.A., S.L.P., C.G., H.M., B.V., L.Y., A.D.R., D.M., N.B., J.G., K.F., C.K., Y.B.B., A.B., V.T., R.P., S.H., and A.B.B. assembled skeletal material, contextual information and provided site descriptions. C.-C.W., S.R., and W.H. analysed data. C.J., I.M., S.S., E.B., O.B. provided additional data and methods. W.H., C.-C.W., S.R., S.H., V.T., R.P., T.H., D.R., and J.K. wrote the manuscript with input from all authors.

## Additional information

**Competing interests:** The authors declare no competing interests.

Chuan-Chao Wang[1,2], Sabine Reinhold[3], Alexey Kalmykov[4], Antje Wissgott[1], Guido Brandt[1], Choongwon Jeong[1], Olivia Cheronet [5,6,7], Matthew Ferry[8,9], Eadaoin Harney[8,9,10], Denise Keating[5,7], Swapan Mallick[8,9,11], Nadin Rohland[8,11], Kristin Stewardson[8,9], Anatoly R. Kantorovich[12], Vladimir E. Maslov[13], Vladimira G. Petrenko[13], Vladimir R. Erlikh[14], Biaslan Ch. Atabiev[15], Rabadan G. Magomedov[16], Philipp L. Kohl[17], Kurt W. Alt[18,19,20], Sandra L. Pichler [19], Claudia Gerling[19], Harald Meller[21], Benik Vardanyan[22,23], Larisa Yeganyan[23], Alexey D. Rezepkin[24], Dirk Mariaschk[3], Natalia Berezina [25], Julia Gresky[26], Katharina Fuchs [27], Corina Knipper[28], Stephan Schiffels [1], Elena Balanovska[29,30], Oleg Balanovsky[29,30,31], Iain Mathieson[32], Thomas Higham [33], Yakov B. Berezin[25], Alexandra Buzhilova[25], Viktor Trifonov[34], Ron Pinhasi [35], Andrej B. Belinskij[4], David Reich[8,9,11,36], Svend Hansen[3], Johannes Krause [1,36] & Wolfgang Haak[1,37]

[1]Department of Archaeogenetics, Max-Planck Institute for the Science of Human History, Kahlaische Strasse 10, D-07745 Jena, Germany. [2]Department of Anthropology and Ethnology, Institute of Anthropology, Xiamen University, 361005 Xiamen, China. [3]German Archaeological Institute, Eurasia Department, Im Dol 2-6, D-14195 Berlin, Germany. [4]'Nasledie' Cultural Heritage Unit, 355006 Stavropol, Russia. [5]Earth Institute, University College Dublin, Dublin 4, Ireland. [6]Department of Anthropology, University of Vienna, 1090 Vienna, Austria. [7]School of Archaeology, University College Dublin, Dublin 4, Ireland. [8]Department of Genetics, Harvard Medical School, Boston 02115 MA, USA. [9]Howard Hughes Medical Institute, Harvard Medical School, Boston 02115 MA, USA. [10]Department of Organismic and Evolutionary Biology, Harvard University, Cambridge, MA 02138, USA. [11]Broad Institute of Harvard and MIT, Cambridge, MA 02142, USA. [12]Department of Archaeology, Faculty of History, Lomonosov Moscow State University, Lomonosovsky pr. 27/4, 119192 Moscow, Russia. [13]Institute of Archaeology RAS, Ul. Dm. Ulyanova 19, 117036 Moscow, Russian Federation. [14]State Museum of Oriental Art, 12a Nikitskiy Boulevard, 119019 Moscow, Russian Federation. [15]Ltd. Institute for Caucasus Archaeology, Ul. Katkhanova 30, 361401 Nalchik, Republic Kabardino-Balkaria, Russian Federation. [16]Institute of History, Archaeology and Ethnography DNC RAS, Ul. M. Jaragskogo 75, 367030 Makhachkala, Republic Dagestan, Russian Federation. [17]Department of Anthropology, Wellesley College, Pendleton East 331, 106 Central Street, Wellesley, MA 02481, USA. [18]Danube Private University, A-3500 Krems-Stein, Austria. [19]IPAS—Institute of Prehistory and Archaeological Science, University of Basel, CH-4055 Basel, Switzerland. [20]Department of Biomedical Engineering, University of Basel, CH-4123 Allschwil, Switzerland. [21]State Heritage Museum, Saxony-Anhalt, D-06114 Halle/Saale, Germany. [22]Martin-Luther-Universität, Halle-Wittenberg D-06108, Germany. [23]Shirak Center for Armenological Studies of National Academy of Science RA, Gyumri 3101, Armenia. [24]Institute for the History of Material Culture, Russian Academy of Sciences, Dvortsovaya nab., 18, 191186 Saint-Petersburg, Russia. [25]Research Institute and Museum of Anthropology of Lomonosov Moscow State University, Mokhovaya 11, Moscow 125009, Russia. [26]German Archaeological Institute, Department of Natural Sciences, Im Dol 2-6, D-14195 Berlin, Germany. [27]CRC 1266 "Scales of Transformation", Institut für Ur- und Frühgeschichte, Christian-Albrechts-Universität, Johanna-Mestorf-Straße 2-6, 24118 Kiel, Germany. [28]Curt Engelhorn Center for Archaeometry gGmbH, 68159 Mannheim, Germany. [29]Research Centre for Medical Genetics, Moscow 115478, Russia. [30]Biobank of North Eurasia, Moscow 115201, Russia. [31]Vavilov Institute for General Genetics, Moscow 119991, Russia. [32]Department of Genetics, Perelman School of Medicine, University of Pennsylvania, Philadelphia, PA 19104, USA. [33]Oxford Radiocarbon Accelerator Unit, RLAHA, University of Oxford, Oxford OX13QY, UK. [34]Institute for the History of Material Culture, Russian Academy of Sciences, Dvortsovaya nab.,18, 191186 Saint-Petersburg, Russia. [35]Department of Evolutionary Anthropology, University of Vienna, 1010 Vienna, Austria. [36]Max Planck-Harvard Research Center for the Archaeoscience of the Ancient Mediterranean, Cambridge, MA 02138, USA. [37]School of Biological Sciences, The University of Adelaide, Adelaide 5005, Australia

