## [Peer Review File · Nature Communications]

Reviewer #1 (Remarks to the Author):

This manuscript is focused on the archaeology of “when and where the formation of “Eurasian ‘steppe ancestry’ “ arose. This ancestry is argued to be a mixture of Eastern and Caucasus hunter-gatherers although it is not known whether it was related to a “horizon of cultural innovations in the 4th millennium BCE that subsequently facilitated the advance of pastoral societies likely linked to the dispersal of Indo-European languages”

The manuscript is authored by a very large number of researchers, most of whom are European archaeologists. The early sections of the manuscript are very poorly written, with little linkage even between paragraphs. It reads as if the archaeologist has a very different agenda and certainly they write in a very different style. The genomics group, comprising about 10 researchers have an interest in the use of genome-wide SNP data to trace ancestry. They have co-authored by a large number of similar papers. The paper utilises very standard analytical methods that have been used by the group on many occasions. I see nothing new in this in relation to methods. More importantly I doubt that more than a handful of readers would be interested to wade through the manuscript.

The substantial problem with the introductory text is that many very specific terms and concepts are introduced, with little or no explanation to the general reader. For example, even something as central to the paper as Maykop which are described as “individuals” and “complexes”. And then. “It is well known for its rich burial mounds, especially at the eponymous Maykop site in today’s Adygea, which reflect the rise of a new system of social organization.

I would suggest to the authors that the abstract needs to be re-written to improve clarity and the manuscript text generally should be very much shortened and re-written to achieve considerably more focused. The introductory material is very protracted and written in a very descriptive style – for example, the first paragraphs lack any kind of direction, it is certainly hard to see where it is going – generally the introductory material needs to be at least halved in length.

The problem with this ms is that archaeology readers will not follow the genomics writings and the converse is equally true. I would suggest that the authors spit the text into two papers and publish each of these in discipline-specific journals after considerable re-writing.

The authors suggest that because individuals buried in kurgans in the “North Caucasian piedmont” appear to be more closely related to ancient individuals from regions further south this allows them to draw two “major conclusions”. But these two conclusions each require further testing – see lines

515-616 and 550-551. I think both of the “major conclusions” are tenuous until this further work is done.

Reviewer #2 (Remarks to the Author):

Comments on Wang et al. for Nature Communications.

I reviewed this manuscript and the Supplementary Materials for archaeology and linguistics. I find no problems with either topic. It is an excellent study by a well-known and accomplished team, on a subject of great significance, because it fills a geographic lacuna in ancient DNA studies that was of the utmost importance in the ancient world. I recommend publication with only a few minor suggestions.

Lines 236 and 238 (again on 544 and 549): To describe the components of ancestry in the same sample, EHG & CHG is used on line 236 (544) and EHG & CHG/Iran on line 238 (549). I understand that Iran Neolithic is largely CHG, so distinguishing between them can be difficult. But CHG should have a clear definition and referent, and be used consistently. If CHG/Iran is used to indicate analyses where CHG could be replaced with Iran Neolithic without changing the result (which seems to be the explanation for lines 236 and 238) then that statistical definition of CHG/Iran should be stated and used consistently. If CHG/Iran is used to indicate CHG & an additional Iran Neolithic ancestry component, then that accretional definition should be stated, or if this accretional definition is never intended, then that should be made clear.

Lines 442 and 606-608 : The abstract for this paper highlights this finding as one of the principal discoveries of the project. The Yamnaya population of the Pontic-Caspian steppes, whose Bronze Age migrations east into Asia and west into Europe laid the foundation for modern populations, is here described as having previously undetected Early European Farmer and Western Hunter Gatherer ancestry. Neither component had been recognized in the same samples when previously published. This is indeed a discovery, and I do not (cannot) debate the finding. I only want to note that this is both a discovery and a revision of results previously published in Nature by Haak et al 2015 Allentoft et al 2015 and Mathiesson et al. 2018. I think that the reader should be alerted more clearly to the potential for debate on this topic. I recognize that methods are moving quickly in this

field, and this analysis might simply supplant analyses from three years ago, but I do not know if this is the case.

Line 632: “bi-directional gene flow”: This study finds evidence of gene flow from Globular Amphorae into Yamnaya , but I am not aware of any Globular Amphorae sample that shows gene flow from Yamnaya. As matters stand now, this is unidirectional gene flow, or cite the sample that shows flow in the other direction.

Figure 2: On the right side, the color bar graphs for Yamnaya Samara and Yamnaya Kalmyk do not show any orange color for Anatolian (or Early European) Farmer ancestry. The discovery of Anatolian/Early European Farmer ancestry in these samples is one of the major findings of the study but is not represented graphically in Figure 2, where these samples are represented graphically as they were described previously, as a simple admixture of EHG and CHG/Iran. I realize that these samples are in a column labeled Previously Published, but this is the only graphical representation of these samples which are said in the text to be composed differently than they are presented in Figure 2.

Reviewer #3 (Remarks to the Author):

The present manuscript by Wang et al is presenting a population genetic analysis of ancient DNA data from the Caucasus region. The same team of authors has published similar papers on other regions (e.g. Mittnik et al (2018) “The genetic prehistory of the Baltic Sea region”), and similar to those papers this one addresses a wide range of archaeological and demographic questions from the particular geographic region. The team uses standard analysis tools and I see the main conclusions supported by data and analysis. My review is from a geneticist point of view and I will not comment on the archaeological background. Most of my comments aim at the presentation of the complex results and explanation of the prehistoric processes.

I am not an expert on this but I think one of the most controversial part may be the last paragraph of the discussion with a potential Indo-European homeland south of the Caucasus. This is in contrast to previous suggestions (by the same authors) placing the homeland north of the Caucasus. I do not understand the arguments why this is parsimonious: the Anatolian branch could be a separate migration from north to south and the authors state that data from south Asia suggests that Indo-European languages got there through the steppe. I do not see why this favors a southern homeland.

Presentation:

Title – the current title reads more like the title of a much more comprehensive survey including all of prehistory and answering all open questions. In fact, the authors only focus on ~3000 years of prehistory. I would suggest to choose a more descriptive and humble title (also maybe adding “human” to the title would make it more descriptive).

The manuscript addresses a range of different questions throughout three millennia of Caucasus prehistory. For someone not familiar with the geographical area and/or the archaeological cultures, that can be hard to follow. It is nice that the Discussion section wraps things up and summarizes the main results for readers from different backgrounds. I would still suggest some improvements to Results section and figures to make following the narrative easier:

- Connect the symbols of Figure 2 to the map of Figure 1
- To understand the “tilting cline”, the reader needs to combine information from different (sub)figures. Maybe a separate plot of latitude vs time with admixture proportions as some color scale would help understanding that part.

The analyses rely heavily on qpAdm and qpWave, conducting many tests of various different models which generates a multiple testing problem which makes it hard to interpret the p values. My personal experience is that the estimates of these tools can be highly dependent on the choice of outgroups, and I am wondering if the authors did any tests to find their set of outgroups. Furthermore, I noticed the following sentence in the caption of supplementary table 5: “...qpWave results can be sensitive to which population is first in the list of testing populations...” - That should obviously not be the case. Is this observation new? It sounds like this could be a major bug, potentially having an effect on the results even if changing the order would not change the estimates.

Minor comments:

- In line 213, the authors claim that individuals were filtered for reference bias. This sounds interesting but I was not able to find information on it in SI3 or Data 1.
- I would recommend to not use the term “insurmountable” several times throughout the manuscript (including the abstract), even if the Caucasus served as a barrier for most migrations it is hard to exclude any.
- I think figure 2C may include the LaBrana1 individual twice, once as shotgun, once as SNP capture data.

- Could Han or another East Asians be used instead of Kennewick for the analysis of Steppe Maykop? I think it might sound misleading to say that “Native American (related)” ancestry is found in the Caucasus if this only reflects to the ANE cline.
- Line 297: Correct “ $|Z < -3|$ ”, and generally check the correct notation for absolute values of Z statistics throughout the manuscript
- Line 394: Please correct “...statistically negative values...”
- The authors say on line 483 that parts of the ancestry for Eneolithic steppe come from a CHG related source. In fact (Figure 5), this (unsampled) source only contributes 44% ancestry to CHG while the remaining 56% come from a source without direct contribution to Eneolithic steppe. This seems surprising as previous research has modeled Eneolithic steppe as EHG+CHG.
- Line 546: Why does this have to be directly linked to the Neolithization? Contact across the Caucasus may have existed in small scale before and that could cause a similar pattern as the same gene pools would be involved.
- I think SI6 is reporting Z scores but in most places they are called “worst f-statistics”.
- Why is supplementary figure 7 only showing $K > 6$?

Point-by-point response to reviewers

We thank the three anonymous reviewers for the constructive feedback and suggestions. We have taken all comments and suggestions on board, and have revised our manuscript accordingly. Please find our response below (blue Italics).

Reviewers' comments:

Reviewer #1 (Remarks to the Author):

This manuscript is focused on the archaeology of “when and where the formation of “Eurasian ‘steppe ancestry’ “ arose. This ancestry is argued to be a mixture of Eastern and Caucasus hunter-gatherers although it is not known whether it was related to a “horizon of cultural innovations in the 4th millennium BCE that subsequently facilitated the advance of pastoral societies likely linked to the dispersal of Indo-European languages”

The manuscript is authored by a very large number of researchers, most of whom are European archaeologists. The early sections of the manuscript are very poorly written, with little linkage even between paragraphs. It reads as if the archaeologist has a very different agenda and certainly they write in a very different style. The genomics group, comprising about 10 researchers have an interest in the use of genome-wide SNP data to trace ancestry. They have co-authored by a large number of similar papers. The paper utilises very standard analytical methods that have been used by the group on many occasions. I see nothing new in this in relation to methods. More importantly I doubt that more than a handful of readers would be interested to wade through the manuscript.

While our study uses state-of-the-art methods used widely in the field, we are convinced that it has still a lot to offer, since the Caucasus bridges the regions of early urbanisation in northern Mesopotamia and those of the Eurasian steppes and our findings have important implications for understanding the founding events leading to the formation of ‘steppe ancestry’ and the critical interaction with farming groups prior to the massive expansions in the 3rd millennium BCE. Evidently, our manuscript is of high interest to the broad readership, as seen by the wide reception it has already received. Over the past 4 months it has been viewed more than 12,000 times, and downloaded nearly 5,000 times on bioRxiv.

The substantial problem with the introductory text is that many very specific terms and concepts are introduced, with little or no explanation to the general reader. For example, even something as central to the paper as Maykop which are described as “individuals” and “complexes”. And then. “It is well known for its rich burial mounds, especially at the eponymous Maykop site in today’s Adygea, which reflect the rise of a new system of social organization.

I would suggest to the authors that the abstract needs to be re-written to improve clarity and the manuscript text generally should be very much shortened and re-written to achieve considerably more focused. The introductory material is very protracted and written in a very descriptive style – for example, the first paragraphs lack any kind of direction, it is certainly

hard to see where it is going – generally the introductory material needs to be at least halved in length.

We followed this suggestion and have shortend/revised the introduction. Please note that we have also an extensive Supplementary section on the ecological and archaeological background of these Bronze Age cultures.

The problem with this ms is that archaeology readers will not follow the genomics writings and the converse is equally true. I would suggest that the authors spit the text into two papers and publish each of these in discipline-specific journals after considerable re-writing.

We tend to disagree on this point. We deliberately chose Nature Communications as a journal with broad and multi-disciplinary scope. Many of our genetics papers have been criticized in the past for exactly the opposite, i.e. being too focused on specific aspects and technical details. Here, we address both the archaeology and the genetics/archaeogenetics community, in an attempt to build bridges between the disciplines.

The authors suggest that because individuals buried in kurgans in the “North Caucasian piedmont” appear to be more closely related to ancient individuals from regions further south this allows them to draw two “major conclusions”. But these two conclusions each require further testing – see lines 515-616 and 550-551. I think both of the “major conclusions” are tenuous until this further work is done.

We agree that “major conclusion” is sounding too strong and have changed this to “two main observations”.

Regarding lines 515-516 (now 501-503): We think that this point comes across convincingly by looking at the position of all modern-day North Caucasus populations, which in PCA space fall precisely between the Bronze Age Steppe and Caucasus clusters. We think that it is important to point this out as future direction. As a consequence, we did not engage deliberately in detailed testing of the relationships between Bronze Age and modern-day populations for two reasons: a) this would expand the scope of the paper immensely and b) warrants a detailed separate study including samples from Iron Age to Medieval times. We are actively collecting samples to pursue the latter.

Regarding lines 550-551 (now 538-539): The individuals from Darkveti-Meshoko (Eneolithic Caucasus) are a strong indicator that Neolithic ancestry had reached the northern flanks by the time of the local Eneolithic. How much earlier this had happened is not directly relevant to the developments that followed during the Bronze Age of the Caucasus region. Similar to the situation above, we are actively pursuing this question. However, the archaeological record for older periods is incredibly scarce and we thus only expect limited progress in the near future.

Reviewer #2 (Remarks to the Author):

Comments on Wang et al. for Nature Communications.

I reviewed this manuscript and the Supplementary Materials for archaeology and linguistics. I find no problems with either topic. It is an excellent study by a well-known and accomplished team, on a subject of great significance, because it fills a geographic lacuna in ancient DNA studies that was of the utmost importance in the ancient world. I recommend publication with only a few minor suggestions.

Many thanks for the positive feedback.

Lines 236 and 238 (again on 544 and 549): To describe the components of ancestry in the same sample, EHG & CHG is used on line 236 (544) and EHG & CHG/Iran on line 238 (549). I understand that Iran Neolithic is largely CHG, so distinguishing between them can be difficult. But CHG should have a clear definition and referent, and be used consistently. If CHG/Iran is used to indicate analyses where CHG could be replaced with Iran Neolithic without changing the result (which seems to be the explanation for lines 236 and 238) then that statistical definition of CHG/Iran should be stated and used consistently. If CHG/Iran is used to indicate CHG & an additional Iran Neolithic ancestry component, then that accretional definition should be stated, or if this accretional definition is never intended, then that should be made clear.

Thanks for pointing out the inconsistency in describing CHG and Iran-like ancestry. We agree that Iran Neolithic is largely CHG as shown in qualitative analysis such as ADMIXTURE components, but we can still distinguish them quantitatively in qpWave and qpAdm modeling because of the difference in their proportion of “basal Eurasian” ancestry. We have checked the manuscript for consistent use of the terminology and added clarifying statements where an absolute distinction of the two related ancestries is needed.

Lines 442 and 606-608 : The abstract for this paper highlights this finding as one of the principal discoveries of the project. The Yamnaya population of the Pontic-Caspian steppes, whose Bronze Age migrations east into Asia and west into Europe laid the foundation for modern populations, is here described as having previously undetected Early European Farmer and Western Hunter Gatherer ancestry. Neither component had been recognized in the same samples when previously published. This is indeed a discovery, and I do not (cannot) debate the finding. I only want to note that this is both a discovery and a revision of results previously published in Nature by Haak et al 2015 Allentoft et al 2015 and Mathieson et al. 2018. I think that the reader should be alerted more clearly to the potential for debate on this topic. I recognize that methods are moving quickly in this field, and this analysis might simply supplant analyses from three years ago, but I do not know if this is the case.

We used established methods for the analysis as our group did in many other publications. The previously underappreciated Early European Farmer and Western Hunter Gatherer ancestry in Yamnaya individuals is certainly of key interest and a novel finding of our study. Our new observation related to the formation of Yamnaya steppe ancestry is based on newly reported Eneolithic steppe individuals, which have not been available to the studies cited above. The new Eneolithic steppe individuals do not show this Anatolian Neolithic +WHG ancestry and this provide an ancestral regional baseline. When we used these Eneolithic

individuals as a local substrate, we noticed that Yamnaya and related subsequent groups needed additional, yet subtle ancestry from Anatolia Neolithic +WHG.

Of note, in our original study from 2015, we did not yet have the CHG individuals and modelled the 'steppe ancestry' in Yamnaya individuals using EHG and modern-day Armenians as proxy. As modern-day Armenians do carry Anatolian farmer-related ancestry in addition to Iranian Neolithic ancestry, this explains why the subtle affinity in Yamnaya individuals went unnoticed. Also, Mathieson et al. 2018 used 'Yamnaya' ancestry in its mixed form for analyses and did not model it anew.

Line 632: "bi-directional gene flow": This study finds evidence of gene flow from Globular Amphorae into Yamnaya, but I am not aware of any Globular Amphorae sample that shows gene flow from Yamnaya. As matters stand now, this is unidirectional gene flow, or cite the sample that shows flow in the other direction.

We agree and have corrected the sentence accordingly. However, the mutual influence is visible in archaeology. In addition, a few individuals on the European farmer side do in fact show subtle evidence for 'steppe' ancestry, such as the Tripolye/Trypillia and Varna Eneolithic individuals (Mathieson et al. 2018) who predate the pastoralist expansion in the 3rd millennium BCE.

Figure 2: On the right side, the color bar graphs for Yamnaya Samara and Yamnaya Kalmyk do not show any orange color for Anatolian (or Early European) Farmer ancestry. The discovery of Anatolian/Early European Farmer ancestry in these samples is one of the major findings of the study but is not represented graphically in Figure 2, where these samples are represented graphically as they were described previously, as a simple admixture of EHG and CHG/Iran. I realize that these samples are in a column labeled Previously Published, but this is the only graphical representation of these samples which are said in the text to be composed differently than they are presented in Figure 2.

Many thanks for pointing this out. We are aware that the Yamnaya individuals do not show Anatolian/Early European Farmer ancestry in the qualitative ADMIXTURE analysis. However, ADMIXTURE is heavily influenced by many factors, such as the number of comparative populations, sample sizes, and how many iterative runs (K) are performed, and thus prone to both artifacts and an under/over estimates of ancestry components. It is widely known in the field, that ancestry proportions should not be read literally from ADMIXTURE plots. Of note, in our updated ADMIXTURE, this component is coming out slightly better than had before, but still not assigned well in Russian Yamnaya individuals. However, as before we do detect this qualitative shift towards European farmers on the PCA plot and have confirmed this observation via formal, quantitative qpWave and qpAdm analysis.

Reviewer #3 (Remarks to the Author):

The present manuscript by Wang et al is presenting a population genetic analysis of ancient DNA data from the Caucasus region. The same team of authors has published similar papers on other regions (e.g. Mittnik et al (2018) "The genetic prehistory of the Baltic Sea region"),

and similar to those papers this one addresses a wide range of archaeological and demographic questions from the particular geographic region. The team uses standard analysis tools and I see the main conclusions supported by data and analysis. My review is from a geneticist point of view and I will not comment on the archaeological background. Most of my comments aim at the presentation of the complex results and explanation of the prehistoric processes.

Many thanks for the positive feedback.

I am not an expert on this but I think one of the most controversial part may be the last paragraph of the discussion with a potential Indo-European homeland south of the Caucasus. This is in contrast to previous suggestions (by the same authors) placing the homeland north of the Caucasus. I do not understand the arguments why this is parsimonious: the Anatolian branch could be a separate migration from north to south and the authors state that data from south Asia suggests that Indo-European languages got there through the steppe. I do not see why this favors a southern homeland.

We agree that the last paragraph is most controversial. In our view, the reconciliation of the linguistic tree topologies with the existing genetic results from ancient DNA studies is still problematic, in particular with regards to the early branches of the PIE-tree. For clarification, if the Anatolian branch, which most linguists agree had branched off first, came from the North, i.e. North-Pontic Steppe, then we should also see an increase in steppe ancestry, and here in particular the EHG component, in individuals of the Anatolian Bronze Age and later periods. However, this is not the case, lending weight to the possibility that Anatolian branched off from Proto-Indo-European south of the Caucasus before it has reached the steppe belt from where it expanded west and east along with 'steppe ancestry' as a genetic tracer dye. However, we are aware that we cannot at present solve this issue and have therefore revised this section using more careful and speculative phrasing. Our intention was to raise awareness about these loose ends of the linguistics, genetics and archaeology triangulation.

Presentation:

Title – the current title reads more like the title of a much more comprehensive survey including all of prehistory and answering all open questions. In fact, the authors only focus on ~3000 years of prehistory. I would suggest to chose a more descriptive and humble title (also maybe adding “human” to the title would make it more descriptive).

Many thanks for pointing this out. We agree and suggest an alternative title for consideration to the editorial board, now reading 'Three thousand years of human genetic history in the Greater Caucasus'.

The manuscript addresses a range of different questions throughout three millennia of Caucasus prehistory. For someone not familiar with the geographical area and/or the archaeological cultures, that can be hard to follow. It is nice that the Discussion section wraps things up and summarizes the main results for readers from different backgrounds. I would still suggest some improvements to Results section and figures to make following the

narrative easier:

- Connect the symbols of Figure 2 to the map of Figure 1
- To understand the “tilting cline”, the reader needs to combine information from different (sub)figures. Maybe a separate plot of latitude vs time with admixture proportions as some color scale would help understanding that part.

Many thanks for these suggestions. We are aware of this issue. The main difficulty was that many individuals come from the same kurgan site, which as a result makes the figure rather more convoluted than clear when the respective symbols are added. We have now revised Figures 1 and 2 to link symbols and improve clarity. We also added the ‘tilting cline’ idea by adding admixture clines described in the main text to the PCA plot.

The analyses rely heavily on qpAdm and qpWave, conducting many tests of various different models which generates a multiple testing problem which makes it hard to interpret the p values. My personal experience is that the estimates of these tools can be highly dependent on the choice of outgroups, and I am wondering if the authors did any tests to find their set of outgroups. Furthermore, I noticed the following sentence in the caption of supplementary table 5: “...qpWave results can be sensitive to which population is first in the list of testing populations...” - That should obviously not be the case. Is this observation new? It sounds like this could be a major bug, potentially having an effect on the results even if changing the order would not change the estimates.

Many thanks for raising this issue. We are aware of this and have therefore largely used an established set of outgroups in our modeling, that has also previously been tested and used for populations of the wider region, in particular in Lazaridis et al. 2016, Nature: Mbuti.DG, Ust_Ishim.DG, Kostenki14, MA1, Han.DG, Papuan.DG, Onge.DG, Ethiopia_4500BP.SG, Karitiana.DG, Natufian, and Iran_Ganj_Dareh_Neolithic. For a number of additional tests, we included Villabruna, Vestonice16, and ElMiron in order to achieve sufficient resolution to distinguish the subtle WHG-related signal in Yamnaya individuals.

The qpWave results can be sensitive to the population order if we use ‘allsnps: YES’ in the parameter setting, which is not a bug, because different populations have different numbers of available SNPs that can be used in the computations. We used ‘allsnps: YES’ option and circulated the order of the first population because we wanted to use as many available markers as possible.

Minor comments:

- In line 213, the authors claim that individuals were filtered for reference bias. This sounds interesting but I was not able to find information on it in SI3 or Data 1.

In the very early stage of our sequencing, we found the shallow shot-gun sequenced samples tend to share more alleles with the hg19 reference in the form of due to the very low coverage. We only use captured data of relatively high coverages in our mature paper. We remove the “reference bias” from line 213 since this method is not relevant to our current analysis.

- I would recommend to not use the term “insurmountable” several times throughout the manuscript (including the abstract), even if the Caucasus served as a barrier for most migrations it is hard to exclude any.

We agree and have corrected the respective paragraphs accordingly.

- I think figure 2C may include the LaBranal individual twice, once as shotgun, once as SNP capture data.

Many thanks for spotting this. We corrected this (and other minor errors) in our revised version of Figure 2.

- Could Han or another East Asians be used instead of Kennewick for the analysis of Steppe Maykop? I think it might sound misleading to say that “Native American (related)” ancestry is found in the Caucasus if this only reflects to the ANE cline.

This is indeed an interesting point. We found that the ‘Native American-related’ ancestry found in the Caucasus was not only caused by the ANE cline, because simple two-way models of Steppe Maykop as a mixture of Eneolithic steppe and AG3 or Kennewick do not fit. We currently suspect that the Kennewick related signal is most likely driven by the East Eurasian part of Native American ancestry. However, at the moment we do not find an East Asian population that can be used as unmixed source for the East Eurasian part of the Native American ancestry.

- Line 297: Correct “ $|Z| < -3$ ”, and generally check the correct notation for absolute values of Z statistics throughout the manuscript

Thanks. This is fixed in the revised manuscript.

- Line 394: Please correct “...statistically negative values...”

Thanks. This is fixed in the revised manuscript.

- The authors say on line 483 that parts of the ancestry for Eneolithic steppe come from a CHG related source. In fact (Figure 5), this (unsampled) source only contributes 44% ancestry to CHG while the remaining 56% come from a source without direct contribution to Eneolithic steppe. This seems surprising as previous research has modeled Eneolithic steppe as EHG+CHG.

We agree that this is an interesting point. The difference to previous results is explained by the use of our newly reported Eneolithic Steppe individuals from the North Caucasus region (PG2004, PG2001, IV3002), which genetically more homogeneous than the Eneolithic individuals from the Samara region (grey triangles in Figure 2D). As shown in Figure 5, the remaining 56% contributed to CHG come from a source related to WHG and EHG. However, given the current resolution it remains difficult to distinguish whether the WHG/EHG related

ancestry in Eneolithic steppe was directly from WHG/EHG or was derived as an admixed form directly from CHG.

- Line 546: Why does this have to be directly linked to the Neolithization? Contact across the Caucasus may have existed in small scale before and that could cause a similar pattern as the same gene pools would be involved.

Please see response to reviewer 1 above. We agree that contact across the Caucasus to the northern flanks could have existed before the Neolithic. However, in the absence of Upper Paleolithic or Mesolithic individuals from the piedmont, this also remains speculative. In our particular case, the increase of Anatolian_Neolithic ancestry must postdate the Mesolithic date of 9700 BP from Kotias Klde (Jones et al. 2015), and thus a time-frame that largely overlaps with the Neolithic in Anatolia and the Near East.

- I think SI6 is reporting Z scores but in most places they are called “worst f-statistics”.

Thanks. This is fixed in the revised manuscript.

- Why is supplementary figure 7 only showing $K > 6$?

We had also run $K=2-6$, but for reason of space decided to only show $K > 6$ onwards in Supplementary Figure 7 because we deemed the detailed structure beyond the continental distinction to be more relevant for question related to West Eurasia. We now show $K=2-18$ in Supplementary Figure 1.

Reviewer #2 (Remarks to the Author):

The authors have addressed all of my comments satisfactorily.

However, they have also added new material and I have seen small problems in old material that I missed before. None of these things are big problems, some are mere typos, and do not change my recommendation to publish, but they should be looked at.

I attempted to upload a word file with these additional comments but it failed to load. Comments pasted below:

Line 135 “later endorsed by” should be “accompanied by”. The date for the domestication of horses cited here in reference 17, 3500 BC, is not “later” than oldest Yamnaya, it is earlier.

Line 145

Insert appropriate abbreviation (EEF) after “Early European farmers”

Line 391

“ranging from Eneolithic steppe to later groups” should be “distinguishing the Eneolithic steppe from later groups”. They describe no Anatolian Farmer in Eneolithic samples, ubiquitous Anatolian Farmer in later samples. There is not a range, but a bifurcation.

Line 533

533 it is possible that Neolithic groups could have reached the northern flanks of the
534 Caucasus earlier⁵⁴ (Supplementary Information 1) and in contact with local hunter
535 gatherers facilitated the exploration of the steppe environment for pastoralist
536 economies.

I suggest deleting the speculation beginning line 534 "and in". Gorelik cited in ref 54 explains the lower Don/Azov steppe Neolithic cultures as a fusion of local steppe foragers with seafaring late-PPNB farmers who, escaping the late-PPNB drought and searching for water, migrated to the northern Anatolian coast (where their sites have not yet been found), then rowed across the Black Sea to the Azov steppes (intervening sites not yet found), possibly also interacting with Mesolithic fishers on the Black Sea coast of the western North Caucasus (sites now under water), but not affecting the northern flanks of the Caucasus (our subject here), which remained Mesolithic. Gorelik did not map any sites with domesticates in the northern flanks of the North Caucasus older in age those reported here, ca 4500 BC, and I am not aware of any that Gorelik missed. It's good to point out that older Neolithic sites could be found in the northern flanks, but what you're implying here is that Gorelik was wrong to specify a PPNB influence on the steppes (fine, I agree) and it really was an extension of the Georgian Shulaveri-Shomutepe culture going over the mountains (sites not yet found) that introduced domesticates to the steppes. Gorelik already dismissed that because of the absence of artifact parallels in the Azov steppes with Shulaveri-Shomutepe. Too complex to explain, too many speculative elements.

Line 618

eastern board > eastern border

Line 655

Armenian, yes, but I don't see why Greek is easily explained geographically by a Caucasus homeland?

Line 640

This hypothesis about the Caucasus source of Proto-Indo-European has been advanced also for slightly other reasons by David Reich and Kristian Kristiansen, so I think it should be elaborated here by the authors and they should marshal their new results to add whatever support they can. However, this hypothesis should rest on showing a sustained admixture between Maikop and Yamnaya to serve as a bridge to Yamnaya from the Caucasus (because the authors accept Yamnaya as connected to later PIE.) It is difficult to see in the results presented here a sustained gene flow from Maikop into Yamnaya, that would sustain this hypothesis. On lines 410 and 432 the authors preferred to see the Anatolian Farmer genes that appeared in Yamnaya as flowing from southeastern Europe, with a 20% WHG component, not from Maikop, without the WHG component. If most of the c. 15% Anatolian Farmer found in Yamnaya came from the west, it leaves very little room for gene flow into Yamnaya from Maikop. If the 3% WHG that makes the difference between a western and Caucasian source of Anatolian Farmer is strongly supported by their data, that makes a Caucasian origin of PIE less likely because it reduces gene flow from Maikop into the steppes. In fact it suggests that very little south-to-north gene flow occurred during the Maikop period (except into 2

individuals from a distinct, small, local genetic group different from Maikop and Yamnaya). This is puzzling and unexpected, but also it fails to support the bridge that seems to be needed.

It did attach, I was wrong.

An important, complex paper that advances new data and conclusions in a field that has been waiting for this. I don't agree with everything said, but I strongly support publishing it, with small corrections.

Reviewer #3 (Remarks to the Author):

This is the second round of reviews of the manuscript by Wang and colleagues on the genetic prehistory of the Greater Caucasus region. I thank the authors for carefully considering the comments made by all reviewers which substantially improved the manuscript. I still have some minor comments (which overlap with my comments from the previous round) that should be included/changed in the manuscript before publication.

I have to admit that I still do not fully understand the issue with the order of populations in qpWave/Adm, but that might be a problem with the limited documentation of these tools. In my understanding, allsnps:YES will cause the software to use all SNPs available for each individual f statistics instead of just the SNPs overlapping between all populations in general (maybe that is already where I am wrong). Then changing the order should not affect the number of SNPs available for an individual f statistic since that number only depends on the populations in that f statistic. It just generally seems odd that a tool would be sensitive to the order of populations in an input file and that this behavior would be considered a feature instead of a bug.

The authors wrote interesting and informative replies to my comments about the Native American ancestry (e.g. "However, at the moment we do not find an East Asian population that can be used as unmixed source for the East Eurasian part of the Native American ancestry."). It would be nice to have such a statement in the manuscript as well. Similar for the response to my comment about the mixed ancestry of CHG in the admixture graph.

Finally, I would like to note that S15 (sorry for writing S16 earlier) still calls Z scores “worst f statistics”.

REVIEWERS' COMMENTS:

Reviewer #2 (Remarks to the Author):

The authors have addressed all of my comments satisfactorily.

However, they have also added new material and I have seen small problems in old material that I missed before. None of these things are big problems, some are mere typos, and do not change my recommendation to publish, but they should be looked at.

I attempted to upload a word file with these additional comments but it failed to load. Comments pasted below:

Line 135 “later endorsed by“ should be “accompanied by”. The date for the domestication of horses cited here in reference 17, 3500 BC, is not “later” than oldest Yamnaya, it is earlier.

Reply: Thanks, this is fixed.

Line 145

Insert appropriate abbreviation (EEF) after “Early European farmers”

Reply: We decided not to insert it as neither ‘EEF’ nor ‘Early European farmers’ is used anywhere else in the manuscript.

Line 391

“ranging from Eneolithic steppe to later groups” should be “distinguishing the Eneolithic steppe from later groups”. They describe no Anatolian Farmer in Eneolithic samples, ubiquitous Anatolian Farmer in later samples. There is not a range, but a bifurcation.

Reply: We agree and have corrected this sentence accordingly.

Line 533

533 it is possible that Neolithic groups could have reached the northern flanks of the
534 Caucasus earlier⁵⁴ (Supplementary Information 1) and in contact with local hunter
535 gatherers facilitated the exploration of the steppe environment for pastoralist
536 economies.

I suggest deleting the speculation beginning line 534 "and in". Gorelik cited in ref 54 explains the lower Don/Azov steppe Neolithic cultures as a fusion of local steppe foragers with seafaring late-PPNB farmers who, escaping the late-PPNB drought and searching for water, migrated to the northern Anatolian coast (where their sites have not yet been found), then rowed across the Black Sea to the Azov steppes (intervening sites not yet found), possibly also interacting with Mesolithic fishers on the Black Sea coast of the western North Caucasus (sites now under water), but not affecting the northern flanks of the Caucasus (our subject here), which remained Mesolithic. Gorelik did not map any sites with domesticates in the northern flanks of the North Caucasus older in age those reported here, ca 4500 BC, and I am not aware of any that Gorelik missed. It's good to point out that older Neolithic sites could be found in the northern flanks, but what you're implying here is that Gorelik was wrong to specify a PPNB influence on the steppes (fine, I agree) and it really was an extension of the Georgian Shulaveri-Shomutepe culture going over the mountains (sites not yet found) that introduced domesticates to the steppes. Gorelik already dismissed that because of the absence of artifact parallels in the Azov steppes with Shulaveri-Shomutepe. Too complex to explain, too many speculative elements.

Reply: We deleted the speculation beginning line 534 "and in" in our revised manuscript.

Line 618

eastern board > eastern border

Reply: Thanks, this is fixed.

Line 655

Armenian, yes, but I don't see why Greek is easily explained geographically by a Caucasus homeland?

Reply: This statement is based on the hypothesis put forward by Mathieson et al. 2018: “An alternative hypothesis is that the homeland of Proto-Indo-European languages was in the Caucasus or in Iran. In this scenario, westward population movement contributed to the dispersal of Anatolian languages, and northward movement and mixture with EHG was responsible for the formation of a ‘Late Proto-Indo European’-speaking population associated with the Yamnaya complex. Although this scenario gains plausibility from our results, it remains possible that Indo-European languages were spread through southeastern Europe into Anatolia without large-scale population movement or admixture.”

Interestingly, the manuscript by Lazaridis et al. 2017, Nature reports additional ‘eastern’ Caucasus-related ancestry in Minoans and Mycenaeans. This also has a bearing on the spread of IE languages to Ancient Greece, in particular since Linear B has been identified as such. As a consequence, and with regards to the direction of the arrival of IE-speakers the authors leave the backdoor open: “Two key questions remain to be addressed by future studies. First, when did the common ‘eastern’ ancestry of both Minoans and Mycenaeans arrive in the Aegean? Second, is the ‘northern’ ancestry in Mycenaeans due to sporadic infiltration of Greece, or to a rapid migration as in Central Europe? Such a migration would support the idea that proto-Greek speakers formed the southern wing of a steppe intrusion of Indo-European speakers. Yet, the absence of ‘northern’ ancestry in the Bronze Age samples from Pisidia, where Indo-European languages were attested in antiquity, casts doubt on this genetic–linguistic association”

Line 640

This hypothesis about the Caucasus source of Proto-Indo-European has been advanced also for slightly other reasons by David Reich and Kristian Kristiansen, so I think it should be elaborated here by the authors and they should marshal their new results to add whatever support they can. However, this hypothesis should rest on showing a sustained admixture between Maikop and Yamnaya to serve as a bridge to Yamnaya from the Caucasus (because the authors accept Yamnaya as connected to later PIE.) It is difficult to see in the results presented here a sustained gene flow from Maikop into Yamnaya, that would sustain this hypothesis. On lines 410 and 432 the authors preferred to see the Anatolian Farmer genes that appeared in Yamnaya as flowing from southeastern Europe, with a 20% WHG component, not from Maikop, without the WHG component. If most of the c. 15% Anatolian Farmer found in Yamnaya came from the west, it leaves very little room for gene flow into Yamnaya from Maikop. If the 3% WHG that makes the difference between a western and Caucasian source of Anatolian Farmer is strongly supported by their data, that makes a Caucasian origin of PIE less likely because it reduces gene flow from Maikop into the steppes. In fact it suggests that very little south-to-north gene flow occurred during the Maikop period (except into 2 individuals from a distinct, small, local genetic group different from Maikop and Yamnaya). This is puzzling and unexpected, but also it fails to support the bridge that seems to be needed.

Reply: We’re afraid that this might be a misunderstanding. There is indeed very limited gene flow between the Caucasus and the steppe groups (apart from the examples highlighted). However, we have based our PIE-related speculations on the observation that the CHG/Iranian (green) ancestry component is increasing already during the Eneolithic north of the Caucasus. This led us to propose that this might be the actual ‘tracer dye’ of an early PIE spread, which could then also accommodate the spread of PIE south of the mountain range where this ancestry component also rises in frequency resulting in a relatively homogenised dual ancestry (Anatolian + Iranian farming-related ancestry) in Chalcolithic times (see also brown arrow in Figure 2).

To emphasise our point, we rephrased the start of the paragraph as follows:

“The insight that the Caucasus mountains served as a corridor for the spread of CHG ancestry north but also for subtle later gene-flow from the south allows speculations on the postulated homelands of Proto-Indo-European (PIE) languages and documented gene-flows that could have carried a consecutive spread of both across West Eurasia^{20, 68}. This also opens up the possibility of a homeland of PIE south of the Caucasus, which itself offers a parsimonious explanation for an early branching off of Anatolian languages, as shown on many PIE tree topologies^{69, 70, 71, 72}.”

It did attach, I was wrong.

An important, complex paper that advances new data and conclusions in a field that has been waiting

for this. I don't agree with everything said, but I strongly support publishing it, with small corrections.

Reviewer #3 (Remarks to the Author):

This is the second round of reviews of the manuscript by Wang and colleagues on the genetic prehistory of the Greater Caucasus region. I thank the authors for carefully considering the comments made by all reviewers which substantially improved the manuscript. I still have some minor comments (which overlap with my comments from the previous round) that should be included/changed in the manuscript before publication.

I have to admit that I still do not fully understand the issue with the order of populations in qpWave/Adm, but that might be a problem with the limited documentation of these tools. In my understanding, allsnps:YES will cause the software to use all SNPs available for each individual f statistics instead of just the SNPs overlapping between all populations in general (maybe that is already where I am wrong). Then changing the order should not affect the number of SNPs available for an individual f statistic since that number only depends on the populations in that f statistic. It just generally seems odd that a tool would be sensitive to the order of populations in an input file and that this behavior would be considered a feature instead of a bug.

Reply: During the analytical steps of this manuscript, we had regular exchange with Nick Patterson, the author of qpWave and qpAdm, who we can quote here:

“qpWave and qpAdm compute f4 statistics of the form $f_4(l_a, l_b; r_a, r_b)$. where l_a, l_b are in a set L , r_a, r_b in a set R . From the trivial identity $(c-d) = (a-d) - (a-c)$, we see that these f_4 statistics have a basis of statistics: $f_4(l_0-l_a; r_0-r_b)$ where l_0, r_0 are fixed.

Thus all relevant f_4 statistics can be written as a linear combination of these basis statistics. The programs take advantage of this. But this is only true for complete data. If there is missing data, as will happen with allsnps: YES then the basis is only approximate. This seems usually to only cause small numerical effects, unless the missing data is somehow biasing the results.

As a result if different base populations are chosen, then the results may change a little. The reviewer is quite correct that this is not ideal, but it is also not easy to fix. A comment is that such dependence on ordering occurs elsewhere in genetics. For example it makes no sense that the likelihood of a set of haplotypes should depend on the order. But for some programs a conditional likelihood of a haplotype is calculated given all “earlier” haplotypes. An example is Matthew Stephens old program PHASE. This is of course not ideal but also not easily fixed.”

The authors wrote interesting and informative replies to my comments about the Native American ancestry (e.g. “However, at the moment we do not find an East Asian population that can be used as unmixed source for the East Eurasian part of the Native American ancestry.”). It would be nice to have such a statement in the manuscript as well. Similar for the response to my comment about the mixed ancestry of CHG in the admixture graph.

Reply: Thanks for your suggestion. We actually have a similar statement in the Discussion section:

“The additional affinity to East Asians suggests that this ancestry does not derive directly from Ancestral North Eurasians but from a yet-to-be-identified ancestral population in north-central Eurasia with a wide distribution between the Caucasus, the Ural Mountains and the Pacific coast, of which we have discovered the so far southwestern-most and also youngest (e.g. the Lola culture individual) genetic representative.”

Finally, I would like to note that SI5 (sorry for writing SI6 earlier) still calls Z scores “worst f statistics”.

Reply: Thanks, this is fixed.